# A large-scale multi-ancestry genome-wide association study of chronic prostatitis/chronic pelvic pain syndrome in men

Sara Brin Rosenthal[1,2], Adam X. Maihofer[2,3], Caroline M. Nievergelt [2,3,4], Daniel Dochtermann [5], Armand Gerstenberger[6,7,8], Thomas Whisenant [1], Saiju Pyarajan [5], Kristina Allen-Brady [9], John N. Krieger[10], Million Veteran Program*, Niloofar Afari[2,3,4,11] & Marianna Gasperi [2,3,4,6,7,8,11] ✉

Chronic prostatitis/chronic pelvic pain syndrome is common, and it impacts men's health and quality of life. The genetic basis of this condition remains largely unknown. Here, we conduct a GWAS using data from the Million Veteran Program of over 590,000 men of European, African, and Hispanic ancestry, including 14,575 chronic prostatitis/chronic pelvic pain syndrome cases. The multi-ancestry analysis identifies eight novel loci associated with chronic prostatitis/chronic pelvic pain syndrome risk, an increase from three significant genome-wide loci found in the European participants alone. We also estimate the genetic correlations between chronic prostatitis/chronic pelvic pain syndrome and 12 phenotypes. Notably, the genetic correlation between chronic prostatitis/chronic pelvic pain syndrome and prostate cancer is not significant. Further, Mendelian randomization shows a significant, potentially bidirectional causal relationship between chronic prostatitis/chronic pelvic pain syndrome and benign prostatic hyperplasia, but not between chronic prostatitis/chronic pelvic pain syndrome and prostate cancer, suggesting a complex interplay between chronic prostatitis/chronic pelvic pain syndrome and benign prostatic hyperplasia. Results of bivariate causal mixture modeling indicate that some of the same genetic variants likely contribute to the development of chronic prostatitis/chronic pelvic pain syndrome, benign prostatic hyperplasia, and prostate cancer.

Chronic Prostatitis/Chronic Pelvic Pain Syndrome (CP/CPPS) is a prevalent and debilitating condition affecting 2–10% of men[1]. Over 90% of all prostatitis diagnoses are CP/CPPS, making it the most prevalent prostatitis diagnosis[2]. Symptoms include persistent pelvic pain and discomfort, sexual dysfunction, and urinary symptoms for at least three months, which can severely impair quality of life[3]. CP/CPPS diagnosis relies on the presence of symptoms of prostatitis in the absence of detectable urinary tract infections and other identifiable

[1]Center for Computational Biology & Bioinformatics, Department of Medicine, University of California San Diego, La Jolla, CA, USA. [2]VA San Diego Healthcare System (VASDHS), San Diego, CA, USA. [3]Department of Psychiatry, University of California San Diego, La Jolla, CA, USA. [4]Center of Excellence for Stress and Mental Health, VASDHS, San Diego, CA, USA. [5]Center for Data and Computational Sciences (C-DACS), VA Boston Healthcare System (VABHS), Boston, MA, USA. [6]VA Puget Sound Health Care System (VAPSHCS), Seattle, WA, USA. [7]Mental Illness Research Education and Clinical Center, VAPSHCS, Seattle, WA, USA. [8]Department of Psychiatry and Behavioral Sciences, University of Washington School of Medicine, Seattle, WA, USA. [9]Division of Epidemiology, Department of Internal Medicine, University of Utah, Salt Lake City, UT, USA. [10]Department of Urology, University of Washington School of Medicine, Seattle, WA, USA. [11]These authors jointly supervised this work: Niloofar Afari, Marianna Gasperi. *A list of authors and their affiliations appears at the end of the paper. ✉e-mail: mgasperi@uw.edu

disorders. This diagnostic approach serves to differentiate CP/CPPS from acute bacterial prostatitis, chronic bacterial prostatitis, and asymptomatic inflammatory prostatitis. Epidemiological studies have linked a history of prostatitis (unspecified, but most symptomatic cases are CP/CPPS) with increased risk for other prostate conditions, including benign prostatic hyperplasia (BPH) and prostate cancer (CaP)[3].

CP/CPPS is characterized by heterogeneity and transient symptomology that can fluctuate over time in intensity and order of appearance. CP/CPPS often mirrors other urological conditions, such as BPH and urological cancers like CaP, as well as chronic overlapping pain conditions (COPCs), complicating diagnosis and leading to inconsistent treatment[3,4]. While the etiology of CP/CPPS is multifaceted and not fully understood, current research suggests a possible pathophysiological overlap with other urinary conditions[5] and implicates prostatic inflammation potentially involving autoimmune responses against prostatic antigens[6] and neurophysiological alternations. A better understanding of the etiology of CP/CPPS is essential to improve diagnosis and therapy.

The extensive comorbidity between CP/CPPS and other COPCs, such as fibromyalgia, cystitis, irritable bowel syndrome, myalgic encephalomyelitis/chronic fatigue syndrome, migraine, tension headache, temporomandibular disorder, and chronic back pain, underscores the nature of CP/CPPS as a chronic pain condition and the potential for shared underlying pathogenic mechanisms[5]. Furthermore, the complex relationship between CP/CPPS and psychiatric conditions, such as post-traumatic stress disorder (PTSD), anxiety, and depression, emphasizes the need for a better etiological model that integrates physiological and psychological aspects of CP/CPPS[7,8]. Understanding the multifactorial nature of CP/CPPS and integrating physiological and psychological factors is likely necessary for comprehensive treatment and management strategies.

Despite heritability estimates from twin studies indicating a substantial genetic component of 36%, the genetic basis of CP/CPPS remains largely uncharted, and the genetic architecture of this condition remains poorly understood[5,9]. Genetic research has not fully utilized genome-wide association study (GWAS) methodologies to evaluate genetic factors contributing to CP/CPPS, which is essential for developing a more complete etiological framework and advancing toward precision medicine. Previous research has shown that genetic factors may contribute to the comorbidity between CP/CPPS and other urinary, COPC, and psychiatric disorders[5,9]. Further clarifying these genetic associations may enhance our understanding of the etiology of CP/CPPS as well as the broader genetic factors involved in male urogenital disorders. The multifaceted etiology of CP/CPPS, documented heritability, and overlap with other conditions present a compelling case to use GWAS to identify genetic factors associated with these conditions.

Our aims were to (1) elucidate the genetic architecture of CP/CPPS by conducting a GWAS within the Million Veteran Program (MVP), a large population-based sample of US Veterans, and (2) explore the associations between CP/CPPS and comorbid urinary and psychiatric conditions. Using the rich genetic and clinical MVP sample, this study was uniquely positioned to explore the genetic nature of CP/CPPS[10] and improve our understanding of its genetic architecture.

In this work, we conduct a GWAS using data from the MVP of over 590,000 Veterans of European, African, and Hispanic ancestry, including 14,575 CP/CPPS cases. In the multi-ancestry meta-analysis, we identify three additional novel loci associated with CP/CPPS risk, in addition to the three novel significant genome-wide loci found in the European cohort alone. We report significant genetic correlations between CP/CPPS and 12 phenotypes, including prostate cancer (CaP), benign prostatic hyperplasia (BPH), genitourinary disease, abdominal and pelvic pain, back pain, depression, and anxiety. Notably, the genetic correlation between CP/CPPS and CaP was insignificant. We identify a significant, potentially causal relationship between CP/CPPS and BPH, but not between CP/CPPS and CaP. This work lays the foundation for future studies to improve our understanding of the biological mechanisms underlying CP/CPPS and to develop new strategies for prevention and treatment.

## Results

Phenotypic prevalence over the available observation period in MVP was 2.2% (22,290/962,151) for CP/CPPS, 3.3% (31,715/962,151) for at CaP, and 31.9% (306,799/962,151) for BPH. Among Veterans with CP/CPPS, 76.26% also had BPH, and 7.59% had CaP. Although many individuals with CP/CPPS also had BPH, most individuals with BPH did not have CP/CPPS. Specifically, among individuals with BPH, only 5.5% had CP/CPPS, and 4.2% had CaP. Among CaP cases, 5.3% had CP/CPPS, and 40.6% had BPH. Tetrachoric correlations showed positive associations, strongest between CP/CPPS and BPH ($r_{tet} = 0.47$ [95% CI = 0.47–0.47, $p < 0.001$]), followed by CP/CPPS and CaP ($r_{tet} = 0.18$ [95% CI = 0.18–0.18, $p < 0.001$]), and CaP and BPH ($r_{tet} = 0.11$ [95% CI = 0.10–0.11, $p < 0.001$]).

The genetic sample included 583,395 participants with 14,575 CP/CPPS cases and 568,820 controls stratified by three HARE race/ancestry groups, with EUR comprising 73.8% of the sample, followed by AFR (18.2%), and HIS (8.1%). Table 1 shows the average age and CP/CPPS prevalence by HARE groups. CP/CPPS prevalence for the three groups was 2.5% and varied across HARE groups, with 2.3% of EUR (10,035 of 430,306 individuals), 3.4% of AFR (3553 of 106,081 individuals), and 2.1% of HIS (987 of 47,008 individuals). Average age varied across the HARE groups from 64.64 (SD = 13.35) years for EUR to 55.84 (15.63) years for HIS, and was higher for CP/CPPS cases than controls in all ancestries ($p < 0.001$). Prevalence rate did not differ between EUR and HIS ancestry ($p = 0.002$).

**Table 1 | CP/CPPS genome-wide association sample composition by HARE-based ancestry and ethnicity**

| Ancestry group[a] | CP/CPPS cases | CP/CPPS controls | Total | P value[b] |
|---|---|---|---|---|
| European (%) | 10,035 (2.33%) | 420,271 (97.67%) | 430,306 (100.00%) | |
| Age (SD) | 67.85 (9.99) | 64.57 (13.41) | 64.64 (13.35) | <0.00001 |
| African | 3,553 (3.35%) | 102,528 (96.65%) | 106,081 (100.00%) | |
| Age (SD) | 63.71 (9.17) | 58.76 (12.02) | 58.92 (11.97) | <0.00001 |
| Hispanic (%) | 987 (2.10%) | 46,021 (97.90%) | 47,008 (100.00%) | |
| Age (SD) | 64.52 (11.20) | 55.65 (15.65) | 55.84 (15.63) | <0.00001 |
| Total | 14,575 (100.00%) | 568,820 (100.00%) | 583,395 (100.00%) | |

[a]Ancestry based on MVP HARE (harmonized ancestry and race/ethnicity) estimates.
[b]P values based on t-test, two-sided.
Prevalence rate was significantly different ($p < 0.001$) for all contrasts except EUR vs. HIS ($p = 0.002$).

**Table 2 | Meta-Ancestry Genome-Wide Significant Loci**

| SNP | Chr | Pos | A1 | A2 | A1 Freq (EUR) | OR (META) | P (META) | Protein coding genes mapped to locus |
|---|---|---|---|---|---|---|---|---|
| rs465498 | 5 | 1325688 | G | A | 0.438 | 0.932 | 7.55E–09 | NKD2, TERT, CLPTM1L |
| rs10886893[a] | 10 | 121284938 | C | T | 0.232 | 1.113 | 1.68E–12 | FGFR2 |
| rs6481483 | 10 | 27818485 | C | T | 0.827 | 1.093 | 9.74E–09 | PTCHD3, MKX, ARMC4 |
| rs157165 | 13 | 50508808 | T | C | 0.229 | 1.090 | 3.61E–09 | PHF11, RCBTB1, EBPL, DLEU1, DLEU7, RNASEH2B, PGLYRP2 |
| rs11084596[a] | 19 | 31614073 | C | T | 0.390 | 0.917 | 2.95E–11 | NA |
| rs62113214[a] | 19 | 50859281 | G | T | 0.075 | 0.806 | 1.72E–14 | LRRC4B, GPR32, C19orf48, KLK15, KLK3, KLK2, AC037199.1, KLK4, KLK7, CTU1, IGLON5, CLDND2, NKG7 |
| rs5969749 | 23 | 16861493 | C | T | 0.4039 | 1.058 | 1.783E–10 | GRPR, MAGEB17, CTPS2, SYAP1, TXLNG, RBBP7, REPS2 |
| rs9698636 | 23 | 17829899 | A | C | 0.0479 | 0.894 | 4.138E–09 | SCML1, RAI2, SCML2 |

*SNP* single nucleotide polymorphism, *Chr* chromosome, *Pos* base pair position on chromosome (GRCh38 Human Genome Build), *A1* Allele 1 (coded), *A2* Allele 2, *A1 Freq* frequency in EUR population, *OR* odds ratio.
[a]Indicates locus was significant in EUR GWAS as well as in the meta-ancestry analysis. Regression estimates tested using a two-sided *z*-test, with exact *P* values in Supplementary Data 2.

## CP/CPPS genome-wide significant loci and genes

The CP/CPPS GWAS included 10,035 cases and 420,271 controls of EUR cohort. Three loci were identified with genome-wide significant associations with CP/CPPS ($p < 5 \times 10^{-8}$) in EUR, including one locus on chr10, a large intergenic region mapping to the gene *FGFR2*, and two loci on chr19 (Table 2, Supplementary Data 1) (Fig. 1A, B). One chr19 locus did not map to any nearby gene, but the other chr19 locus encompassed a large coding region that included 13 mapped genes (*LRRC4B, GPR32, C19orf48, KLK15, KLK3, KLK2, AC037199.1, KLK4, KLK7, CTU1, IGLON5, CLDND2, NKG7*). Of these genes, *KLK2* and *KLK3* were mapped by both physical proximity (by being near the lead SNP) and eQTL associations, *KLK15* and *AC037199*.1 were mapped by physical proximity only, and *LRRC4B, GPR32, C19orf48, KLK4, KLK7, CTU1, IGLON5, CLDND2*, and *NKG7* were mapped by eQTL association only (Fig. 1A, B and Table 1). MAGMA was also run on the summary statistics as an alternative gene-mapping strategy. The genes identified by MAGMA were consistent with FUMA (Supplementary Data 2). Of 118 SNPs significantly associated with CP/CPPS in the three GWS loci ($p < 5 \times 10^{-8}$), 26 had a regulomeDB score of 3a or lower, indicating possible regulatory relationships (Supplementary Data 3). Conducting the analyses with an age of MVP-enrollment covariate yielded the same three genome-wide significant loci and lead SNPs. The genetic correlation between the EUR CP/CPPS GWAS without age and the EUR CP/CPPS GWAS was 0.9922 (SE = 0.181).

Tissue-specific enrichment analyses with MAGMA revealed significant enrichment for the prostate ($p = 2 \times 10^{-4}$) and bladder ($p = 0.02$), indicating that genes associated with CP/CPPS are specifically expressed in relevant tissues (Supplementary Data 4). MAGMA pathway analysis reveals significant enrichment for brain-related pathways (Supplementary Data 5), including negative regulation of neurotransmitter secretion, Purkinje system development, brain morphogenesis, and midbrain development.

SNPs in the three genome-wide significant loci had prior associations with relevant phenotypes, including CaP and prostate-specific antigen (PSA) levels, through the GWAS catalog. The chr19 locus, which did not map to any nearby gene, had a known association with BPH, lower urinary tract symptoms, and PSA[11,12]. (Fig. 1D and Supplementary Data 6).

An analysis of marginally associated SNPs ($p < 5 \times 10^{-6}$) with FUMA revealed an additional 18 loci, on top of the 3 already found, in which 73 genes were mapped (Fig. 1A). These marginally mapped genes were strongly enriched for prostate (Supplementary Fig. 1), using the TSEA tool[13]. TSEA is a method similar to MAGMA's tissue-specific enrichment analysis, but is designed for the input of lists of significant genes, as opposed to MAGMA's strategy of collapsing the GWAS signal within the gene set of interest. We used TSEA to evaluate if the genes mapping

from the marginal FUMA analysis likely contained true positives previously missed by the stringent threshold. Furthermore, SNPs within the marginal loci had previous associations with relevant phenotypes, including CaP, BPH, lower urinary tract symptoms, and PSA levels (Fig. 1E)[14]. This strong enrichment for relevant tissues and previous associations with relevant phenotypes implies that some marginally associated genes represent true positive associations with CP/CPPS.

## SNP-based heritability

Our SNP-based heritability analysis of EUR CP/CPPS identified a mean $C^2$ of 1.0815, and we observed a modest genomic control inflation factor GC $\lambda = 1.0736$, suggesting that our results are unlikely to be influenced by population stratification or other sources of bias. An LDSC intercept of 1.0235 (SE: 0.0074) and attenuation ratio of 0.2883 (SE: 0.0905) indicated that additive genetic effects primarily drive the heritability of CP/CPPS. The observed scale EUR $h^2$ was estimated at 6.7% (SE: 0.12) with a corresponding Z-score of 5.43, indicating that genetic factors explain a portion of the variance in CP/CPPS.

## Multi-ancestry meta-analysis

To increase power to detect smaller effect sizes, multi-ancestry meta-analysis was conducted on summary statistics from CP/CPPS GWAS on EUR (10,035 cases, 420,271 controls), AFR (3,553 cases, 102,528 controls), and HIS (987 cases, 46,021 controls). The multi-ancestry analyses (14,575 cases, 568,820 controls) yielded three additional autosomal genome-wide significant loci on chr5, chr10, and chr13 (Supplementary Fig. 2; Table 2 and Supplementary Data 1) and two loci on chr23 (X). The chr5 locus mapped to 3 genes: *NKD2, TERT*, and *CLPTM1L*. SNPs in this locus have previously been associated with PSA levels and BPH in GWAS studies[11]. The new chr10 locus mapped to 3 genes: *PTCHD3, MKX, ARMC4*. SNPs in this locus have previously been associated with PSA levels and BPH[11,15]. The chr13 locus mapped to 6 genes, including *DLEU1, EBPL*, and *PHF11*. SNPs in this locus have previously been associated with PSA levels and BPH[11,12]. Two out of the three GWS loci newly identified in the multi-ancestry meta-analysis were also identified in the marginal SNPs EUR analysis ($p < 5 \times 10^{-6}$), providing more evidence that some of the marginally significant loci do contain relevant information. The two chr23 loci mapped to 10 genes (Table 2), of which *GRPR* is involved in spinal itch signaling, *REPS2* is associated with androgen receptor signaling, and *RAI2* is linked to higher PSA levels[16].

We also examined the results of individual ancestry GWAS. While the AFR cohort did not yield any significant loci, one genome-wide significant SNP (CHR2:52764460; rs144707855) was found in the HIS cohort that was found neither in the EUR cohort nor the multi-ancestry analysis, indicating that it may be a HIS-specific variant.

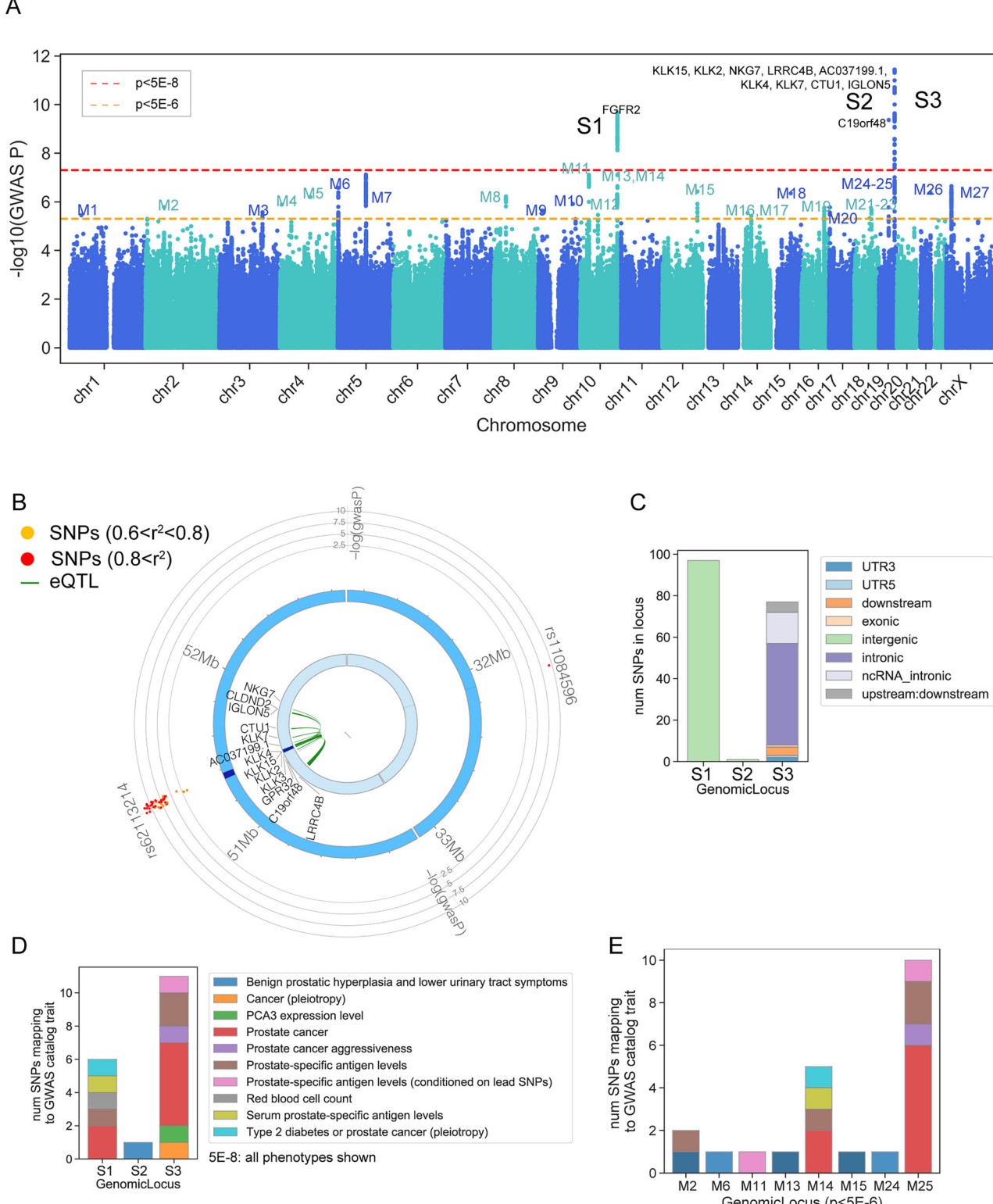

**Fig. 1 | Summary of significant loci (EUR HARE race/ancestry only). A** Manhattan plot showing negative log p-value of the European-only CP/CPPS GWAS. $N = 430,306$ (10,035 cases, 420,271 controls). Mapped genes from FUMA are annotated. S1-S3 refer to the genome-wide significant loci ($p < 5 \times 10^{-8}$ = red dotted line). M1-M27 refer to the marginal loci ($p < 5 \times 10^{-6}$ = yellow dotted line). Regression estimates tested using a two-sided z-test, with exact *p*-values in Supplementary

Data 2. **B** Circos plot for the two Chr19 loci. **C** The barplot shows the composition of SNP type for significant SNPs per GWS locus. **D** Barplot showing the number of SNPs in each GWS locus ($p < 5 \times 10^{-8}$) previously associated with traits from the GWAS catalog. All associations are shown. **E** Barplot showing the number of SNPs in each marginal GWS locus ($p < 5 \times 10^{-6}$) previously associated with CP/CPPS-relevant traits.

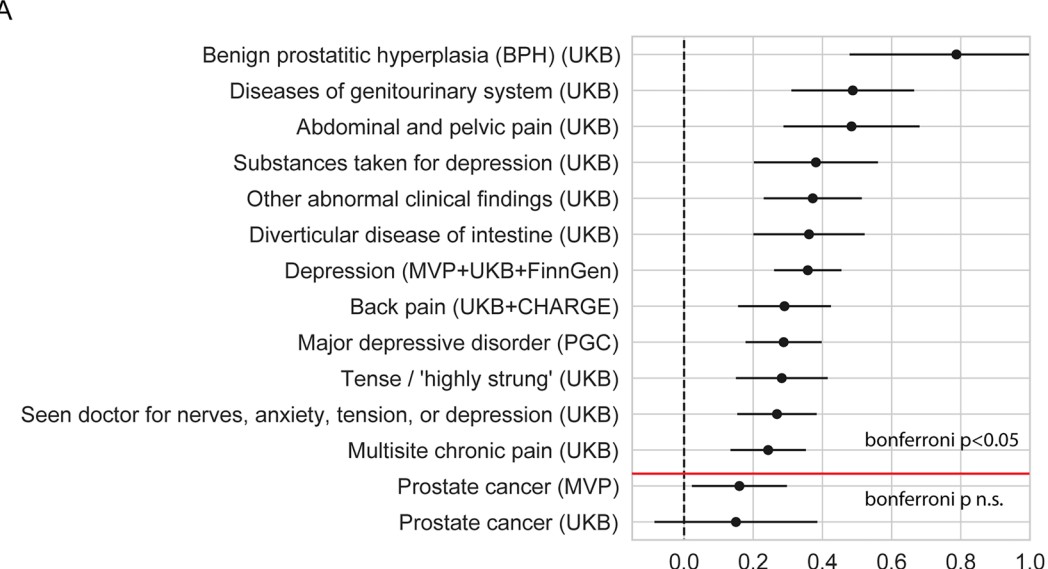

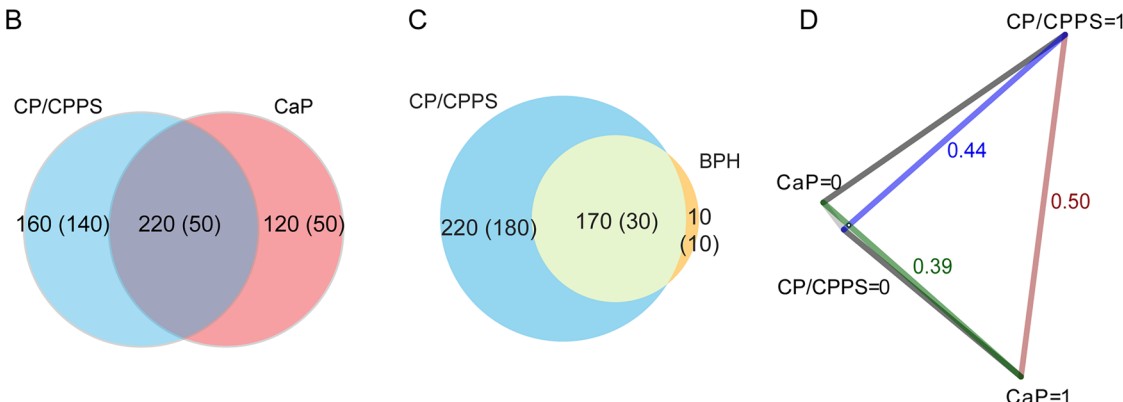

**Fig. 2 | Relationship to other traits. A** LDSC genetic correlation between CP/CPPS EUR GWAS (10,035 cases, 420,271 controls) and traits that meet Bonferroni $p < 0.05$ (above the red line), along with CaP from MVP and UKB. The figure shows rg (circle) with 95% confidence intervals. P values are found in Supplementary Data 7. **B** Venn diagram depicting the degree to which CP/CPPS and CaP share genetic origins from MiXeR. We observe that CP/CPPS and CaP have genetic overlap (220 predicted causal variants), but there is also variation specific to either disease (CaP: 120 predicted causal variants, CP/CPPS: 160 predicted causal variants). The standard error is shown parenthetically. P-values are displayed in Supplementary Data 8.

**C** Venn diagram depicting the degree to which CP/CPPS and BPH share genetic origins from MiXeR. P values are shown in Supplementary Data 8. We observed that CP/CPPS and BPH have genetic overlap (170 predicted causal variants), variation specific to CP/CPPS (220 predicted causal variants), and less, which is specific to BPH (10 predicted causal variants). The standard error is shown parenthetically. **D** Genetic distance between CP/CPPS cases (CP/CPPS = 1), CaP cases (CaP=1), CP/CPPS controls (CP/CPPS = 0), and CaP controls (CaP = 0) from case-case GWAS analysis.

## Relationship to other traits

To gain further insights into the genetic basis of CP/CPPS, we conducted an unbiased LDSC screen of genetic correlations with 1466 publicly available GWAS using the complex traits genetics virtual lab (CTG-VL)[17]. These data include phenotypes from the UK Biobank, GIANT consortium, Psychiatric Genomics Consortium, FinnGen, and CHARGE. These analyses allowed us to identify significant genetic correlations between CP/CPPS and a range of other phenotypes (with a stringent Bonferroni-corrected $p < 0.05$; Supplementary Data 7). We observed significant genetic correlations between CP/CPPS and 12 other phenotypes. Consistent with other studies[5], among these were genitourinary and pain conditions, BPH, diseases of the genitourinary system, and abdominal and pelvic pain (from UKB) (Fig. 2A). Other significantly correlated phenotypes were related to depression, anxiety, and non-urinary pain, including back pain and multisite pain (from

UKB, MVP, and PGC, among others; Fig. 2A). These results confirm previous work that suggested CP/CPPS and other COPCs may have a shared genetic component[9,18,19]. These findings support previous suggestions of the urological and psychiatric relationships of CP/CPPS[18,20,21] and the interconnected nature of chronic pain and mental health.

The FUMA analysis of GWAS Catalog findings for CP/CPPS revealed associations between the lead SNPs in this study and a range of prostate-related traits, including CaP, BPH, serum PSA levels, and CaP aggressiveness. Full GWAS Catalog results are presented in Supplementary Data 6.

## Relationship to BPH and CaP

Genetic correlation analysis with LDSC revealed that CP/CPPS and CaP had a moderate positive but not significant post Bonferroni-correction

correlation ($r = 0.16$, SE: 0.07). Interestingly, the magnitude of correlation was much lower than that observed for BPH (rg = 0.78, SE = 0.16) and depression and pain phenotypes (Fig. 2A).

As an alternative approach, we quantified the degree to which CP/CPPS and CaP and CP/CPPS and BPH share genetic origins using MiXeR[22]. From the bivariate MiXeR analysis (Supplementary Data 8), we observed polygenic overlap, defined as the ratio of causal variants associated with both traits divided by the causal variants associated with either trait, as 0.44 (Fig. 2B) for CP/CPPS and CaP and 0.43 for CP/CPPS and BPH, indicating that some of the same genetic variants contribute to the development of all three conditions. Between CP/CPPS and CaP we observe variation specific to each trait, while between CP/CPPS and BPH, there is much less variation specific to BPH (AIC > 0, Fig. 2B, C). In contrast to genetic correlation, which computes the proportion of variance two traits share, the polygenic overlap measures the fraction of predicted causal variants shared by both traits.

Mendelian randomization (MR) was used to evaluate the potential causal relationships between CP/CPPS, BPH, and CaP, since these phenotypes often appear together. We modeled CP/CPPS as the exposure variable and BPH and CaP as the outcome variables using MRbase[23]. All selected SNPs had F-statistics greater than 10, confirming the strength of the instrument.

When MR was applied to CP/CPPS and BPH, we found mixed results across different methods. The IVW method, weighted median method, and simple mode methods all indicated a significant positive relationship between CP/CPPS and BPH (IVW: $b = 0.041$, SE = 0.013, $p = 0.002$; Weighted median: $b = 0.036$, SE = 0.005, $p < 0.001$; Simple mode: $b = 0.057$, SE = 0.010, $p = 0.032$) (Supplementary Data 9a). Neither the weighted mode method nor the MR Egger method resulted in a significant association ($p = 0.099$, $p = 0.427$, respectively). The MR Egger intercept term was not significantly different from zero ($-0.0074$, $p = 0.14$), suggesting no evidence of directional pleiotropy. Thus, three out of five methods tested supported a potential causal relationship between CP/CPPS and an increased risk of BPH. Intriguingly, MR also yielded significant results when the exposure and outcome variables were reversed (with BPH causing CP/CPPS) in the analysis of CP/CPPS and BPH (Supplementary Data 9b), with all five methods tested yielding a significant positive association between BPH and CP/CPPS ($p < 0.05$). The MR Egger intercept term was not significantly different from zero (0.0863, $p = 0.11$), suggesting no evidence of directional pleiotropy. Because BPH primarily affects the prostate gland and is characterized by its enlargement, BPH may contribute to the development of CP/CPPS. These significant bidirectional results from MR suggest a complex interplay between CP/CPPS and BPH that should be explored further with better-powered samples and additional variants.

As expected, based on the non-significant correlation between CP/CPPS and CaP, when MR was applied to CP/CPPS and CaP, we did not observe a significant effect with any of the five methods ($p > 0.05$), suggesting that CP/CPPS is not a causative factor for CaP (Supplementary Data 9c).

CAUSE analyses yielded similar results when comparing causal and sharing models (Supplementary Data 10). For CP/CPPS against BPH, the causal model was a better fit than both the null ($p = 0.02$) and sharing models ($p = 0.01$), suggesting a causal relationship from CP/CPPS to BPH. Similarly, for BPH against CP/CPPS, the causal model was a better fit than the null ($p = 0.01$) and sharing models ($p = 0.03$), suggesting a causal relationship in the reverse direction. These bidirectional results likely indicate shared genetic architecture or pleiotropy rather than distinct causal mechanisms in both directions. When CAUSE was applied to CP/CPPS and CaP, neither the sharing nor the causal model was significant, suggesting that horizontal pleiotropy may explain the association between these two phenotypes. Various models of comorbidity can exist, and it is possible that causality

operates in both directions, necessitating more research to understand these relationships fully.

## Case-case GWAS

As an alternative approach to identify CP/CPPS-specific risk loci, we performed a case-case GWAS (CC-GWAS)[24], comparing CP/CPPS cases to CaP cases. Case-case GWAS is a computational method that compares the allele frequencies between two different case-control cohorts and operates on pre-computed summary statistics. CC-GWAS identified 1113 GWS SNPs with significantly different allele frequencies in cases of CP/CPPS versus CaP cases. Of these, 1112 were also GWS SNPs in CaP, none were GWS in CP/CPPS, and 1 was uniquely identified by CC-GWAS (rs8113621). Additionally, two loci were identified within these 1113 SNPs, suggesting potential regions of interest that may be associated with the genetic distinction between CP/CPPS and CaP cases. The first locus on chromosome 11 (lead rs7126851) included *SIGIRR*, *TOLLIP*, and *BRSK2*, which are involved in immune response and cellular signaling pathways, as well as other genes (Supplementary Data 11). The second locus (lead rs11650165) on chromosome 17 included *AC007461.1*. The finding that most CC-GWAS-identified SNPs are also significant in CaP indicates that variants that distinguish CaP from controls also distinguish CaP from CP/CPPS and that the phenotypes are at least partially driven by different genetic origins. Consistent with this observation, CaP and CP/CPPS cases have a CC-GWAS genetic distance value ($F_{ST}$) of 0.50. This value is nearly equivalent to the genetic distance between CP/CPPS cases and controls ($F_{ST} = 0.44$) (Fig. 2D). A comparative analysis with BPH was not feasible due to the high correlation between CP/CPPS and BPH that limits our ability to distinguish between the genetic factors influencing each condition.

## Proximal interaction network

Conducting network analyses of the potentially underpowered CP/CPPS GWAS enables the integration of diverse datasets, enhancing detection of marginal signals that may have been overlooked. Network analysis can uncover highly interacting genes, providing insights into potential molecular pathways and mechanisms that underlie CP/CPPS, ultimately contributing to a better understanding and improved treatment.

We built an integrated CP/CPPS molecular interaction network using the technique of network propagation[25], a tool useful for integrating datasets and boosting marginal signals. Nineteen (19) high confidence CP/CPPS genes from the multi-ancestry analysis out of the original 28 were contained within the STRING background interactome (9 genes had no interaction information in STRING: *AC037199.1*, *C19orf48*, *CLDND2*, *DLEU1*, *DLEU7*, *EBPL*, *PGLYRP2*, *PTCHD3*, and *VSIG10L*). Following propagation seeded by the 19 high-confidence CP/CPPS genes in STRING, we identified 102 highly proximal genes in the network. These included the 19 high-confidence CP/CPPS genes themselves, two medium-confidence FUMA genes (*KLK1*, *NKX3*-1), and 81 network-proximal genes (Fig. 3).

Functional enrichment analyses across a wide array of curated pathways and gene sets of the 38 CP/CPPS-network genes revealed significant enrichment for genes downregulated in the aging prostate (adj. $p = 1 \times 10^{-8}$, from GTEx aging signatures, Fig. 3B). We also identified a significant association with androgen receptor signaling that has been linked to CaP (adj $p = 5 \times 10^{-9}$, Fig. 3B)[26].

Patients with CP/CPPS are at a higher risk for other neurological conditions, including migraine headaches, internalizing psychopathology, depression, and anxiety[9]. Interestingly, the CP/CPPS network was significantly enriched for brain-related terms and pathways, including synapse interactions, organization, and neuronal system in general (Supplementary Data 12 and Fig. 3B). The genes driving this brain-specific enrichment include *LRRC4B*, *PTPRF*, *NTNG1*, *DNER*, and *LRRC4*. *LRRC4B* is of note, as it is a high-confidence CP/CPPS GWAS

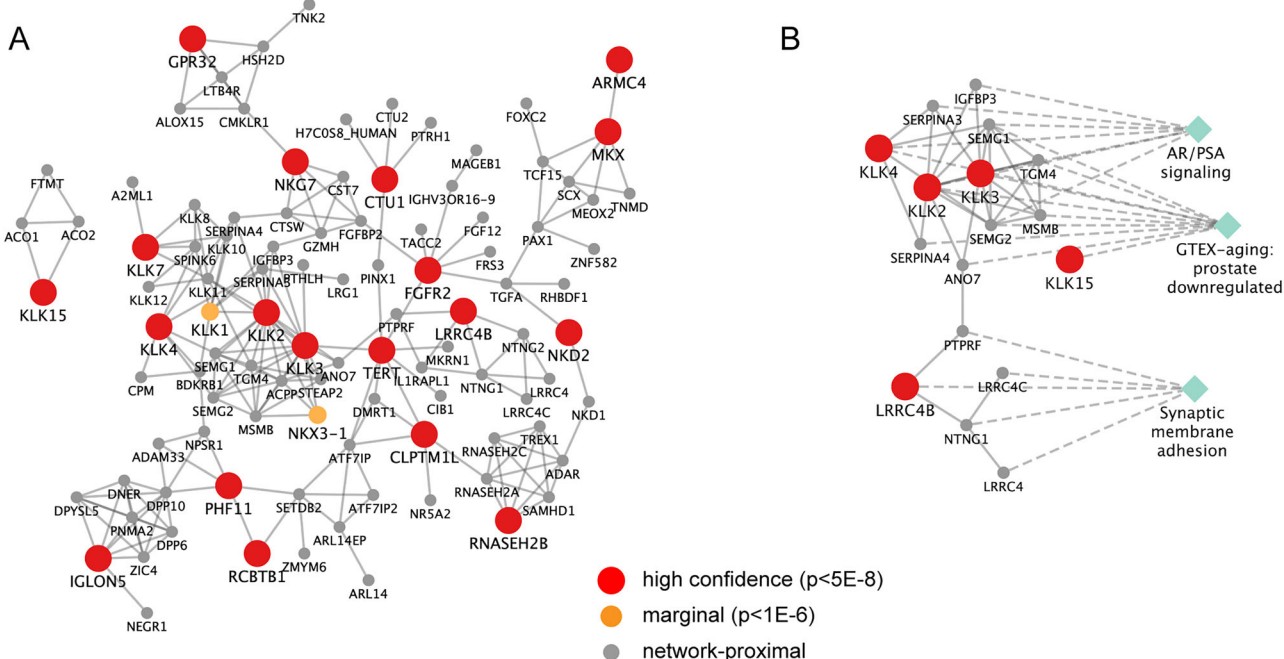

**Fig. 3 | CP/CPPS-proximal Network. A** Network of genes that are significantly proximal to the genes mapped to GWAS loci ($z > 4$). Genes mapped to GWS loci ($p < 5 \times 10^{-8}$) are shown with red circles, while genes mapped to the marginal loci ($p < 5 \times 10^{-6}$) are shown with orange circles, and genes that were not mapped to any CP/CPPS locus but were proximal in network space are shown with gray circles. Edges indicate an interaction from STRING. To define 'significantly proximal' genes in the network, we used a permutation-based empirical test (as implemented in Rosenthal et al., 2023). **B** Subgraph showing genes from the CP/CPPS-proximal network, which are significantly enriched for aging prostate in GTEx, androgen receptor/PSA signaling, and synapse organization. Green diamonds indicate pathways, and a dotted line indicates the genes in each pathway. *P* values are in Supplementary Data 12. Enrichr uses a one-sided Fisher's exact test (right-tailed) for functional enrichment (to see whether the input gene list is enriched for that term more than expected by chance). We report both nominal and adjusted *p* values in Supplementary Data 12. The pathways reported here are all significant after multiple tests (adj $p < 0.05$).

gene, is also involved in synaptic processes, and is predicted to play an important role in the postsynaptic density membrane. These brain-related pathways in the CP/CPPS network could underpin the observed neurological comorbidities.

## Replication

As a source of replication, we queried the FinnGen trait called "Inflammatory disease of the prostate (prostatitis)," where the cases were coded using the presence of ICD-10 code N41, a non-specific parent code that includes various conditions involving the prostate (acute prostatitis, chronic prostatitis, abscess of prostate, prostato-cystitis, granulomatous prostatitis, other inflammatory diseases of prostate, and inflammatory disease of prostate, unspecified) rather than specifically CP/CPPS. These data contained 4160 cases and 130,139 controls. In the summary statistics for this trait, the lead SNPs for two out of three genome-wide significant loci replicated at nominal significance (rs10886893: $p = 0.03$, and rs62113214: $p = 0.001$). The genome-wide significant SNPs from the muti-ancestry analysis did not replicate in FinnGen ($p > 0.05$).

A search of publicly available transcriptomics data on CP/CPPS within GEO yielded limited results. We identified one study, GSE11842, that generated gene expression data from the prostate epithelium and ventral prostate tissue from control and vinclozolin-treated rats[27]. After multiple generations, the treated animals presented symptoms that model prostate disease. While standard differential expression analysis comparing control and treated rats identified 0 genes with adjusted *p* value < 0.05, a targeted approach focusing on the members of the CP/CPPS-proximal interaction network identified 5 genes with a nominally significant *p* value < 0.01. *FGFR2* is decreased with treatment in both tissue types and was identified above as significantly associated with CP/CPPS. KRT5 is increased with treatment and is a marker for the basal epithelium in prostate tissue[28]. *IGHMBP2*, *PPFIBP1*, and *CCDC59* are increased in both tissues with treatment, but no known evidence for their role in the prostate. *IGHMBP2* is an immunoglobulin mu DNA-binding protein, *CCDC59* is a coiled-coil domain-containing protein indirectly involved in RNA binding, and *PPFIBP1* has been associated with various non-prostate-related tumors. These results support the GWAS results as they relate to *FGFR2* and suggest additional genes like *KRT5* for further investigation.

## Discussion

In this GWAS of CP/CPPS, we identified genetic variants associated with this complex condition, estimated its heritability, and conducted functional analyses to understand its genetic architecture and the relationship between CP/CPPS and comorbid conditions. We found several novel loci associated with CP/CPPS risk and provided insights into its genetic basis. Our findings resonate with earlier biometric twin studies[5,9], suggesting the contribution of genetic factors in CP/CPPS and supporting the presence of genetic overlap with other urinary disorders, such as BPH and CaP. Furthermore, our results suggest that some of these genetic variants may play a role in developing other comorbid conditions, including BPH, CaP, depression, and other COPCs.

CP/CPPS GWAS with the European samples identified 3 genetic loci. These loci are known to be related to CaP and PSA levels. Expanding our CP/CPPS GWAS in a multi-ancestry analysis revealed five additional genome-wide significant loci. Three of these additional multi-ancestry loci were marginally significant in the GWAS from European samples alone, demonstrating the potential boost in power the additional multi-ancestry samples provided. We note that the additional loci are not ancestry-specific but rather were identified from the additional power afforded by the larger sample size resulting from

including an increased number of EUR admixed individuals. Our findings suggest that genetic factors contribute to CP/CPPS and highlight the need to further investigate the underlying genetic mechanisms of this condition.

While previous efforts by our group using the classical twin design estimated the heritability of CP/CPPS at 36%[9], our EUR SNP-based heritability estimate was 6.7%. The "missing heritability" discrepancy may be due to the modest sample size and power of the current CP/CPPS GWAS, as well as the differences between GWAS and biometric twin modeling methodologies. More specifically, because heritability estimates from GWAS are derived from common SNPs, they may overlook other contributing factors, including rare variants, structural variants, epistasis, or gene-environment interactions that are captured in twin studies. The difference highlights the complexity of CP/CPPS genetics and the need for integrating diverse methodologies to improve our understanding. Similarly, the LDSC-based genetic correlations between CP/CPPS and BPH (rg = 0.78, SE = 0.16) and CaP (rg = 0.16, SE = 0.07) in this study were lower than our previous twin-study estimates of rg = 0.87 (0.67–0.91) and 0.48 (0.33–0.99), respectively[9]. Both sets of results point to a genetic relationship between these conditions. Differences in the magnitude of the genetic correlations can be attributed to the methodologies and limitations of the two approaches, sample differences, and statistical power.

Epidemiological studies suggest relationships between a history of "prostatitis" (which usually represents CP/CPPS), CaP, and BPH, but the details remain unclear[1,29,30]. In particular, a 5-fold increase in CaP, along with increases in BPH and lower urinary tract symptoms, has been observed in men who reported CP/CPPS symptoms[1]. Genetic correlations with BPH and CaP highlight distinct yet overlapping genetic components, with bidirectional relationships suggesting complex underlying biological mechanisms. The strong genetic correlation between CP/CPPS and BPH, along with GWAS catalog associations, suggests that the underlying genetic factors may be similar for these two conditions, a finding supported by the bidirectional relationship between CP/CPPS and BPH indicated by MR. However, this pattern may reflect shared genetic etiology or pleiotropy rather than strictly causal effects in both directions. Several possible scenarios may underlie these results, including common genetic factors influencing both conditions, pleiotropy where the variants influence both CP/CPPS and BHP through distinct biological pathways, methodological factors like phenotype heterogeneity, true bidirectional causality, or some combination. Consistent with our other results, the finding suggests that factors contributing to the development of one condition might also increase the risk of the other and vice versa, underscoring the need to improve our understanding of the biological mechanisms and pathways shared between these conditions. While clinical comorbidity between CP/CPPS and BPH may contribute to the observed relationship, the moderate phenotypic correlation in our sample, together with genetic findings from two independent approaches (LDSC-derived correlations and twin biometric modeling), support the presence of shared genetic architecture between these two conditions, potentially involving inflammatory processes and endocrine changes.

In contrast to the strong genetic link between CP/CPPS and BPH, the association between CP/CPPS and CaP appears to be more modest. Although CP/CPPS and CaP have a high comorbidity, the observed genetic correlation in our study was low. This suggests that while the two conditions share some genetic influences, the observed comorbidity may be due to environmental factors, including clinical practice. Our MR findings were consistent with this, indicating that while CP/CPPS and CaP share some genetic predisposition, there is genetic variation specific to CP/CPPS that is not shared by the two conditions. Understanding shared genetic factors between CP/CPPS and these conditions may lead to improved diagnostic methods and therapeutic interventions.

Our insights into the CP/CPPS association with PSA provide additional context for understanding this condition. PSA, a glycoprotein enzyme encoded by *KLK3*, is a biomarker used to detect and monitor CaP, as CaP can disrupt normal prostate function and increase PSA levels. High PSA levels are correlated with cancer severity, but since PSA levels can be impacted by other factors, including inflammation[31], age, genetics, and the presence of BPH[32], it does not have a high predictive value for CaP diagnosis[33]. Thus, the fact that genome-wide significant SNPs from our CP/CPPS phenotype have also been associated with PSA does not imply that they also impact CaP. In other words, CaP and PSA are correlated but distinct phenotypes. Notably, while this chr19 locus did have known associations with relevant phenotypes, it has not been previously associated with CaP[12]. This finding is also supported by the association of *RAI2* with PSA levels.

Our findings extend beyond CP/CPS to suggest neurobiological links with pain-related and psychiatric conditions, underscoring the genetic complexity and broader health implications. The tissue-specific enrichment analysis results are important given the established relationship between CP/CPPS, depression, and non-urological COPCs[9]. Enrichment for brain-related pathways in CP/CPPS suggests a potential neurobiological link underlying these relationships. Chronic inflammation may potentially influence central pain processing pathways and contribute to pain perception and mood dysregulation. Understanding these neurobiological mechanisms could be critical in understanding the pathophysiology of CP/CPPS and its neuropsychiatric comorbidities, and possibly leading to more effective treatment strategies for both sets of conditions.

Importantly, the *FGFR2* locus, associated with prostate pathology and brain signaling, shows the potential of genes to mediate CP/CPPS and its comorbidities, warranting further investigation. In particular, the upregulation of fibroblast growth factors (FGFs) is associated with promoting angiogenesis and cell proliferation, involved in prostate enlargement that may be characteristic of BPH[34]. Furthermore, FGFs are critical signaling molecules in the brain, with roles in development, maintenance, and repair[35]. Thus, it is possible that the *FGFR2* locus could contribute to the observed comorbidity between CP/CPPS and BPH. Further work is needed to elucidate this relationship.

Our study revealed genetic links between CP/CPPS and various comorbid conditions, including COPCs such as back pain and abdominal pain, as well as mental health conditions, including depression and anxiety. Consistent with this pathway and network analyses of CP/CPPS, GWAS revealed significant enrichment in brain-related pathways, suggesting that genetic causes of CP/CPPS may be related to pain sensation and perception. These findings align with prior research and highlight the interplay and genetic overlap between CP/CPPS, COPCs[7,9,19], and psychiatric conditions, underscoring the potential for improved diagnostic and treatment approaches through the understanding of these genetic associations.

The functional enrichment finding revealing significant enrichment for genes downregulated in the aging prostate and association with androgen signaling could point to a possible connection between the aging process of the prostate and the development of CP/CPPS. Downregulation of these genes might contribute to an increased susceptibility to CP/CPPS or exacerbate symptoms.

Our study has important limitations. First, although this CP/CPPS GWAS study included multiple ancestry groups, people of East Asian and South Asian ancestry groups within the MVP cohort were small and, therefore, not analyzed. Heritability estimates were conducted in the EUR sample, where a sufficiently large sample size was available to produce reliable estimates. This limitation highlights the need to recruit larger and more diverse cohorts. Another limitation was the absence of external replication data. Because no other GWAS of CP/CPPS have been conducted, we could not replicate our findings using independent samples from other sources. The absence of replication

datasets within other biobanks, including UKB and All of US, highlights the need for collaborative efforts and the collection of additional CP/CPPS data. Assessment of CP/CPPS was primarily based on EHR and ICD codes. Such data are valuable for large-scale studies. However, EHR-based assessment may not capture the full spectrum of clinical manifestations and may be subject to inconsistencies in coding and documentation. Because CP/CPPS is a complex and heterogeneous condition with various clinical presentations, it is unlikely that all cases of CP/CPPS are adequately reflected in ICD codes. We were not able to validate the ICD codes using chart review, which may reduce the statistical power of our findings. However, despite this limitation, the strong associations observed in our study indicate that our results are still robust and warrant further investigation. EHR may also be incomplete because patients may seek care elsewhere or have not yet developed the conditions. Additionally, post-GWAS analyses were limited to autosomal chromosomes, as most methods and summary statistics currently do not include the X chromosome. Finally, although we observed a bidirectional MR relationship between CP/CPPS and BHP, these findings may be more consistent with shared genetic factors or pleiotropy rather than a causal relationship.

Future research, including thorough clinical assessment, will be needed to further clarify the genetic interplay of these connections.

In summary, our work advances the growing understanding of the genetic architecture of CP/CPPS and provides a foundation for future genetic studies. We identified six loci in the multi-ancestry cohort associated with CP/CPPS, including ones involved in prostate development and non-malignant prostate pathology that will be important to investigate in future studies. Our findings also shed light on the genetic relationship of CP/CPPS to CaP and BPH and support the genetic contribution to the overlap between CP/CPPS and psychiatric conditions like depression and other COPCs. These findings highlight how genetic studies can inform the clinical management of CP/CPPS, supporting the future development of personalized, genetically-informed treatment options.

## Methods

### Ethics
The VA Central Institutional Review Board (cIRB) approved the MVP protocol in 2010, and enrollment began in 2011. The VA cIRB and the Research and Development Committee at the VA San Diego Healthcare System and VA Puget Sound Health Care System approved these analyses (under project "MVP033").

### Study participants and design
Observational data were from the MVP, a national research project to determine how genetic traits, health habits, and environmental factors affect Veteran health and illness. MVP's design and recruitment procedures have been previously documented[10]. In brief, MVP data include self-reported surveys, electronic health records (EHR), and genetic data. MVP Veterans were included if they had complete data. At the time of analysis (May 2023), 819,417 Veterans were enrolled in the MVP.

### Race, ethnicity, and ancestry
We used Harmonizing Genetic Ancestry and Self-identified Race/Ethnicity (HARE) groups to define race/ethnicity[36]. Briefly, HARE enhances classification by integrating self-identified race/ethnicity (SIRE) and genetically inferred ancestry (GIA). HARE improves the accuracy of classification using GIA to refine and, if necessary, impute SIRE, improving reliability of race/ethnicity assignment in genetic research. Less than 2% of individuals are not assigned a HARE group when participant-identified and genetically inferred ancestry data produce discordant results. Here, we use the term "Hispanic" (HIS) for the HARE race and ethnicity groups comprised of individuals who are Latino or Hispanic, the term "European" (EUR) for individuals who are White but

not Hispanic, and "African" (AFR) for individuals who are Black but not Hispanic. People of East Asian and South Asian ancestry were not analyzed due to the low numbers in MVP. Hereafter, we refer to the HARE groups used in analyses as HIS, EUR, and AFR.

### GWAS cohort and phenotype
Of 819,417 enrolled Veterans, individuals were excluded if they did not have sufficient EHR records ($n = 121$), were identified as women ($n = 75,911$), missing gender ($n = 131$), were not genotyped ($n = 144,119$), had missing HARE race/ancestry estimates ($n = 8516$), or had a small HARE race/ancestry group ($n = 7355$). Overall, we analyzed data from 583,395 individuals.

A total of 583,395 participants had available phenotype and genotype information and were used for GWAS analysis (Table 1). The reference population groups (EUR, AFR, HIS) in the 1000 Genomes samples were used to define EUR, AFR, HIS ancestries for CP/CPPS, and EUR for CaP. SNPs with an imputation INFO score > 0.3, minor allele frequency (MAF) ≥ 0.01, and HWE > $1 \times 10^{-15}$ were reported in the analysis. For the primary analysis, a genome-wide significance was set as a $p$ value ≤ $5.0 \times 10^{-8}$.

### Chronic prostatitis/chronic pelvic pain syndrome (CP/CPPS)
CP/CPPS cases and controls were identified using EHR ICD 9/10 codes to generate a binary variable. ICD codes 601.1 and N41.1 were used to identify CP/CPPS cases, defined as Veterans with one or more of the ICD codes. Controls were defined as Veterans who did not have either ICD code.

### Genotyping, imputation, and quality control
Genotyping, imputation, and quality control within MVP have been described previously and were conducted by MVP Core[10]. Briefly, MVP samples were genotyped using a 723,305-SNP Affymetrix Axiom Biobank array, customized to include variants of interest in multiple diverse ancestries[10]. Imputation was performed with Minimac4 using data from the TopMed reference panel. Analyses were performed using MVP Release 4 data (GRCh38). Final genotype data consisted of 96 million variants. Principal components were calculated for each ancestry using PLINK 2.0 alpha[37].

### Computation and statistical analyses
**GWAS regression and meta-analysis.** GWAS analysis was carried out separately for CP/CPPS in EUR, AFR, and HIS. Logistic regression analyses were performed to test association between phenotypes and imputed dosages using REGENIE v3.1.3. GWAS was performed on populations stratified by HARE ancestry. The model included the first 10 principal components of genotype as covariates. A sensitivity analysis was carried out with an additional covariate of age of MVP enrollment. SNPs with imputation INFO scores > 0.3, minor allele frequency (MAF) ≥ 0.01, and HWE > $1 \times 10^{-15}$ were reported. A genome-wide significance (GWS) was set for the primary analysis as $P ≤ 5.0 \times 10^{-8}$. Post-GWAS analyses were conducted utilizing autosomal SNPs, as most methods rely on autosomal data, and public summary statistics frequently do not include the X chromosome.

REGENIE is a two-step machine learning method that adjusts for relatedness. Step One analyzed MVP genotype array data only, dividing SNPs into blocks and using ridge regression to generate predictions. These predictions were combined in second ridge regression and decomposed by chromosome for leave-one-chromosome-out analysis, which served as covariates in Step Two. Step Two used Release 4 imputed data for cross-validation and applied Firth logistic regression and saddle point approximation for the binary trait analysis.

Meta-analyses were conducted across summary statistics from 3 ancestries (EUR, AFR, and HIS) using Plink[37] and MR-MEGA[38], with default parameters. Results were consistent across both platforms.

The 1000 Genomes phase 3 reference data[39] were used to calculate LD, relying on ancestry-specific reference genotypes.

**Functional annotation with FUMA.** SNPs were annotated to nearby and relevant genes using the FUMA tool, positional mapping, and eQTL mapping from all current eQTL databases available on FUMA. We excluded old versions of genotype-tissue expression (GTEx) still available on the platform[40]. Default settings were used in all FUMA analyses unless specified. The SNP2Gene module was used to identify independent genomic risk loci and variants in LD with lead SNPs ($r^2 > 0.6$, calculated using ancestry-appropriate 1000 Genomes reference: EUR for European Ancestry, AFR for African ancestry, and AMR for Hispanic Ancestry). FUMA results are reported in hg38. Both positional and eQTL information were used for gene mapping. FUMA was run twice, once with stringent parameters (high confidence genes; $p < 5 \times 10^{-8}$) and once with relaxed parameters (medium confidence genes; $p < 5 \times 10^{-6}$).

**Gene-based and gene set, and tissue-enrichment analyses with MAGMA.** We used the MAGMA[41] tool, analyzed through the FUMA webtool, for gene-based, gene-pathway, and tissue enrichment analyses.

**Tissue-specific enrichment analysis (TSEA).** Tissue-specific enrichment analysis on the genes mapped to marginally significant EUR loci ($p < 5 \times 10^{-6}$) was run using the TSEA tool[13].

**Additional regulatory information.** SNPs with potential regulatory relationships were further annotated through regulomeDB[42], to identify genes known to bind to focal sites through ChIP assays.

**Linkage Disequilibrium Score Regression and SNP-Based Heritability.** Linkage disequilibrium score regression (LDSC)[43] was used to estimate SNP-based heritability of CP/CPPS. A CP/CPPS prevalence of 8.2% was used[1,44,45]. The 1000 Genomes phase 3 reference data (1KGPp3)[39] were used to calculate LD. The extent to which test statistic (GC λ) was due to polygenic signal (rather than population stratification) was calculated with LDSC as 1 - (LDSC intercept - 1) / (mean observed $\chi^2$ - 1)[43].

**Genetic correlations with traits.** We evaluated the genetic relationship of CP/CPPS with BPH and CaP. BPH summary statistics from the UK BioBank (UKB)[46] were used. BPH was defined by UKB-based self-report (data field 20002_1516: Non-cancer illness code, self-reported: bph / benign prostatic hypertrophy). CaP summary statistics were derived from the MVP based on an EHR ICD 9/10 codes (185, V10.46, C61, Z85.45, Z85.46, Z85.47, Z85.49, and Z85.9) with patients considered cases with at least four ICD codes for CaP to account for routine billing of diagnostic biopsies. Controls were defined as having no ICD codes for CaP.

To further explore the relationship between CP/CPPS and other phenotypes, we leveraged publicly available GWAS data within the Complex Trait Genetics Virtual Lab[17] (https://vl.genoma.io/). Cross-trait LDSC regression was performed on various conditions to assess genetic correlation with CP/CPPS. To ensure robust findings, we restricted our analyses from the 1437 available phenotypes to those displaying a SNP heritability z-score greater than 4, yielding 823 phenotypes for evaluation. Bonferroni adjustment was applied to control for multiple comparisons, setting the threshold for significance at $p < 0.05/823 = 6.08 \times 10^{-5}$.

**Mendelian randomization.** Mendelian randomization (MR) was performed using the R package associated with MRbase, TwoSampleMR[23]. We modeled CP/CPPS as the exposure variable and BPH and CaP as the outcome variables. To evaluate potential bidirectional relationships, we also conducted MR analyses using BPH and CaP as the exposure and CP/CPPS as the outcome. Before running the MR, we ran clumping and pruned for LD as recommended by MRbase. MR was performed using 5 common methods (MR Egger, Weighted median, Inverse variance weighted [IVW], Simple mode, and Weighted mode). To ensure the strength of our instrumental variables (IVs), we calculated the F-statistic using the formula $F = \beta^2/(SE)^2$ and considered SNPs with F-statistic greater than 10. The variance explained by each SNP was calculated using $\beta^2 \times 2f \times (1 - f)$ where $f$ is MAF. SNPs in high LD ($r^2 \geq 0.001$) with a more significant SNP within a 1 Mb window were pruned. Harmonization of exposure and outcome data was performed to ensure allele alignment and consistency using the "harmonize_data" function. To ensure independence, SNPs were screened for associations with potential confounders using the GWAS Catalog and by evaluating the MR Egger intercept to assess the presence of directional pleiotropy. MR-Egger intercept test was used to assess the robustness of results and address potential pleiotropy. Finally, we used CAUSE, an MR approach using full GWAS summary statistics to evaluate the causal relationship of CP/CPPS on BPH and CP/CPPS on CaP[47]. To avoid false positives that can occur using other methods, CAUSE models correlated and uncorrelated horizontal pleiotropy.

**Mixed effects score regression (MiXeR).** We employed the mixed effects score regression (MiXeR) framework[22] to estimate genetic overlap between CP/CPPS and CaP and between CP/CPPS and BPH. MiXeR utilizes a Bayesian approach to provide posterior probabilities for two key estimates: the number of shared and trait-specific loci. This approach facilitates unbiased estimation of genetic overlap, independent of the power of individual GWAS. Analyses were conducted using MiXeR v1.3.

**Case-case GWAS (CC-GWAS).** To further investigate shared genetic architecture between CP/CPS, CaP, and BPH, we utilized a case-case GWAS (CC-GWAS) approach[24]. This method integrates summary statistics from GWAS summary results for each condition, leveraging the combined power across results to identify genetic loci contributing to more than one trait and to detect cross-phenotypic associations. The implementation of CC-GWAS focused on identifying loci that exhibit pleiotropic effects that are common between CP/CPPS and CaP and BPH.

**Integrative network analysis.** We employed a network propagation algorithm[48] to connect the multi-ancestry high-confidence genes (genes annotated to $p < 5 \times 10^{-8}$ loci) to any interacting medium-confidence genes (genes annotated to $p < 5 \times 10^{-6}$ multi-ancestry loci), thus, boosting the signal from any marginal results. The network propagation was seeded with the CP/CPPS high-confidence FUMA-mapped genes using the STRING high-confidence interactome (v12)[49]. Highly significantly interacting genes ($z > 4$) were retained for further analysis. Functional enrichment analysis was calculated with enrichR[50].

**Replication.** A search of the Gene Expression Omnibus (GEO) using the term "prostatitis" identified one gene expression study (GSE11842) that compared control and treated prostate epithelium and ventral prostate tissue in a transgenerational model of vinclozolin-induced prostate disease[27]. Differential gene expression was performed using limma with the model (-tissue type + treatment)[51]. Other search results identified studies that were excluded due to tissue type (i.e., colonic epithelium, cecal bacteria) or measurement target (i.e., miRNA, methylated DNA).

FinnGen (https://www.finngen.fi/en) trait called "Inflammatory disease of the prostate (prostatitis)" (Release 10, 4,160 cases, 130,139 controls) was also compared with the EUR results from the current study.

**Reporting summary**

Further information on research design is available in the Nature Portfolio Reporting Summary linked to this article.

## Data availability

MVP summary data are available via dbGaP under accession phs001672. Access requires submission of a data access request; raw individual-level data are protected and not publicly available due to privacy regulations. For replication, FinnGen release 10 summary statistics for the phenotype "Inflammatory disease of the prostate" (prostatitis; 4,160 cases, 130,139 controls) were used as the best available analog, though CP/CPPS is a distinct phenotype. The phenotype description and GWAS overview are available at https://r10.finngen.fi/pheno/N14_PROSTATITIS. Full summary statistics for release 10 (finngen_R10_N14_PROSTATITIS.gz) or later releases are available upon registration via the FinnGen portal: https://elomake.helsinki.fi/lomakkeet/124935/lomake.htm.

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

## Acknowledgements

Nievergelt is funded by the Department of Veterans Affairs, Biomedical Laboratory Research and Development (BLRD) BX005920, and Rehabilitation Research and Development (RR&D) RX004293. Gasperi is supported by the VA Career Development Award #1IK2CX002107 from the US Department of Veterans Affairs Clinical Science Research and Development Service. This research was partially supported by the Altman Clinical & Translational Research Institute (ACTRI) at the University of California, San Diego. The ACTRI is funded by awards issued by the National Center for Advancing Translational Sciences, NIH UL1TR001442. The authors gratefully acknowledge the continued cooperation and participation of the members of the Million Veteran Program; without their contribution, this research would not have been possible. This research is based on data from the Million Veteran Program (Project MVP033), Office of Research and Development, Veterans Health Administration. This publication does not represent the views of the Department of Veteran Affairs or the United States Government.

## Author contributions

Obtained funding: M.G. Clinical: M.G., N.A., J.K., and K.A.B., Statistical analysis: S.B.R., M.G., A.X.M., A.G., T.W., D.D., S.P., and C.M.N., Writing group: S.B.R., M.G., A.G., T.W., N.A., J.K., K.A.B., and C.M.N.

## Competing interests

The authors declare no competing interests.

## Additional information

## Million Veteran Program

Adam X. Maihofer[2,3], Caroline M. Nievergelt[2,3,4,7], Daniel Dochtermann [5], Armand Gerstenberger[6,7,8], Saiju Pyarajan [5], Niloofar Afari[2,3,4,11] & Marianna Gasperi [2,3,4,6,7,8,11] ✉

