## [Transparent Peer Review file · Nature Communications]

A large-scale multi-ancestry genome-wide association study of chronic prostatitis/chronic pelvic pain syndrome in men

Corresponding Author: Dr Marianna Gasperi

Version 0:

Reviewer comments:

Reviewer #1

(Remarks to the Author)

Review of:

The Genetic Basis of Chronic Prostatitis/Chronic Pelvic Pain Syndrome: Insights from a Large-Scale Multi-ancestry Study

This study addresses an important biomedical question, specifically the common genetics between Chronic Prostatitis/Chronic Pelvic Pain Syndrome and other conditions affecting the prostate and other conditions generally. The study is based on the Million Veteran Program study including 590,000 veterans of European, African and Hispanic ancestry from the United States, of which 14,575 have CP/CPPS. The most interesting findings are lines of evidence supporting common genetics between CP/CPPS and benign prostate hyperplasia but not between CP/CPPS and prostate cancer. Overlapping genetics between CP/CPPS and other traits are also examined highlighting neurological function, matching some well-known findings of overlapping genetics across pain phenotypes and neurological function.

While the overall idea of the study is one that merits strong consideration in terms of a prominent venue for publication, the manuscript in its present form suffers from poor or incomplete execution in several key ways that weaken the whole to the point where a thorough re-working of the manuscript would be necessary. These impact on how convincing or well-supported the conclusions of the study are.

--Combined CP/CPPS phenotype

The numbers of study subjects classified as cases who were attributed an ICD code of one or the other out of CP and/or CPPS phenotypes is not detailed. How many study subjects were attributed codes for both phenotypes? Based on information provided in the first para of the introduction, most who have CP also have CPPS type symptoms. However, what is not detailed is the proportion of individuals with CPPS who have CP. Both sides would need to be examined carefully in a study that now puts the two together presenting these as a single phenotype, and in two ways – via the literature, and via the dataset being used. Basic univariate analyses of the phenotypes considered in the study should be performed. E.g. the reader should be presented with the proportion of CPPS individuals who have benign prostate hyperplasia. What about those who have prostate cancer? What about the distribution of the 12 phenotypes according to CP and CPPS status? It is impossible to tell from the information provided whether the genetic overlap with phenotypes like backpain may be due to a higher proportion of CPPS compared to CP in the combined phenotype. The second introduction paragraph evokes “unpredictable symptomatology” – what is unpredictable: the order of symptoms over time, the magnitude of importance of each element; the number of possible symptoms that must be present for a diagnosis to be positive? Are the authors saying that the diagnosis process for identifying this condition is highly problematic and therefore it isn't clear what CP and or CPPS is anyway? How is the diagnosis made in the clinic? Would there be a lot of disagreement across physicians on attributing a CP or CPPS diagnosis? The starting point of the present study is so fuzzy that none of the results can be interpreted in any meaningful way, it seems.

Mendelian Randomization:

Basic reporting guidelines were not followed. Examination of the four assumptions was not undertaken. 5 methods were tried and one reported in the results section. Going to the table S8, only two methods give a nominally statistically significant result. The robust Egger method is not statistically significant. There is no process for screening the instrumental variables.

What were the instrumental variables used? What about sensitivity analyses? Logic in the paragraph II. 453-465 does not hold up: the sentence “we modeled...” suggests a uni-directional approach was used. Later it is suggested a bi-directional approach was used. This paragraph leaves an impression of confusion. These results simply cannot be interpreted or trusted.

Paragraph II. 610-623 goes into lots of detail about a well know truism relating to common complex disease: twin and family-based study estimates of heritability are always higher than SNP chip heritability. This paragraph adds nothing that is specific to prostate conditions.

Lines 698 etc. Are there no other biobanks that have recorded prostate condition data? If the authors tried to go to other biobanks and searched for other resources with both prostate condition data and genotyping data but the data was not available, this should be detailed explicitly. What needs to be done to remedy this situation so that prostate conditions get adequate representation from biobanks?

Table 1: Why aren't tests presented showing differences in case proportions by ancestry group? Why aren't percentages of the different ancestries as a part of the whole MVP presented?

Line 356: the authors seem to be confused on how to interpret RegulomeDB scores. Lower scores show greatest evidence of regulatory involvement.

Gene symbols are sometimes in italics, sometimes not. Check your manuscript before submitting it.

Reviewer #2

(Remarks to the Author)

1. In the GWAS Cohort and Phenotype section, the term “ancestry” is used. Based on the context, I believe the authors are likely referring to the categorical HARE race variable for MVP participants. I think that moving the Race, Ethnicity, and Ancestry section before the GWAS Cohort and Phenotype section, and stating that the groupings of dropped individuals were based on the HARE race/ancestry categories would perhaps avoid misunderstandings with readers who are not familiar with the HARE approach. In subsequent mentions elsewhere in the paper, I also think it would be prudent to adjust the terminology to specify that you are using the HARE race/ancestry categories to stratify participants into analysis groups. This is clearly presented in Table 1, but I feel the precise language should be used throughout the paper. For example, then sentence: “The CP/CPPS GWAS included 10,035 cases and 420,271 controls of European ancestry.” On line 343, would be more accurately stated: “The CP/CPPS GWAS included 10,035 cases and 420,271 controls from the HARE race/ancestry category.” This is because those individuals are predominantly European and self-identify as “non-Hispanic white” but are not necessarily entirely European.
2. The abbreviations EUR, AFR, HIS are established on line 205 without defining them. I think it would be fine to state that these abbreviations refer to the HARE race/ancestry categories and use them on subsequent mentions. In general I see many instances of switching between using these abbreviations and XX ancestry. I think it is important to clean this up and use a consistent naming convention based on the approach.
3. The IGC should be changed to the Greek lambda (line 394)
4. Were all the statistics presented in the SNP-based heritability section derived from analysis of EUR data? Perhaps the HIS was too small, but perhaps the AFR data could provide an estimate.
5. Fst is usually presented with a subscript
6. GTE_x should be spelled out and cited on first mention
7. Is there any measure of how well a single instance of an ICD code classifies either of these outcomes (i.e. a PPV compared to expert review of records)? If not, this should be acknowledged as a limitation and an opportunity for improvement in future studies in the Discussion.

Reviewer #3

(Remarks to the Author)

Rosenthal S.B. et al performed the first GWAS of Chronic Prostatitis/Chronic Pelvic Pain Syndrome (CP/CPPS) in men with 14,575 cases and 568,820 controls without CP/CPPS. They identified a total of six loci across ancestry groups, including a Hispanic ethnicity specific locus. The majority of these loci have been associated with related phenotypes including benign prostatic hyperplasia (BPH) and prostate-specific antigen (PSA), supporting the hypothesis of the multifaceted etiology of CP/CPPS. They follow the GWAS with appropriate methods to further identify pleiotropy between CP/CPPS and other related phenotypes (CaP and BPH). They further demonstrate an interconnection between CP/CPPS and neurological disorders. The work presented supports the knowledge of the field and provides routes of further investigation into the pleiotropy of CP/CPPS and BPH. Overall, the authors performed the study well, they utilized appropriate methods, and the results are novel and clearly communicated.

I have a few comments the authors may wish to consider:

Comments:

1. It would be beneficial to include the X chromosome in the analyses.
2. To further the authors' point of the shared genetic architecture between CP/CPPS and other diseases, it could be

beneficial to conduct a PheWAS of the identified loci (potentially just the clumped and pruned variants utilized in the MR) to investigate the other associated phenotypes.

Minor Comments:

1. While age was not included in the model, it would be worthwhile to conduct a sensitivity analysis including age as a covariate.
2. As the authors use the CC-GWAS method for CaP and BPH, it would be worthwhile to include the MR intercepts as further support of potential pleiotropy between the phenotypes.
3. While informative, Figure 1B is difficult to read at the presented font size.
4. Line 432: Authors state 1,466 traits while the methods state 823 traits.
5. Line 455 – 458: If the authors could clarify this section. Based on the authors description of the exposure (CP/CPPS) and outcome (BPH) and result, it would seem that CP/CPPS may be causative of BPH, however, it is written as BPH may be causative of CP/CPPS.
6. While the title includes prostatitis suggesting a study of men, the title should include the denotation of including only men in the study following the Nature Reporting Summary.

Version 1:

Reviewer comments:

Reviewer #1

(Remarks to the Author)

Abstract :

MiXer is mentioned as a jargony black box rather than in language that can be interpreted by a scientist who has not used the software, especially for the general audience targeted by Nature Communications. The authors should interpret what MiXer contributed without using the acronym MiXer.

Introduction:

Some clarity is still missing from the explanation of the CP/CPPS diagnosis. A sentence like “CP/CPPS is a diagnostic label attributed to chronic prostatitis cases for which a concomitant infection is not detected” is needed here as the provided text does not explain that a diagnostic algorithm is followed by physicians with this label being one possible classification.

“In the multi-ancestry meta-analysis, we identify six novel loci associated with CP/CPPS 158 risk, an increase from three significant genome-wide loci found in the European cohort.”

Are you saying the three loci identified among Europeans are not novel? Does this mean they have been reported elsewhere? Or do you mean there are three that are also found in other ethnicities, showing validation across different ancestries? If they are not novel and have been reported elsewhere, references should be provided.

Under reviewer 1, comment 1, the authors did not take the comment into consideration.

Reviewer 1 comment 3

3) A basic statistical analysis of the phenotypes available to researchers in the MVP cohort should be provided. The authors have chosen to ignore this reviewer comment.

The interpretation of genetic correlations is much more meaningful if this can be done in the light of phenotypic correlations. Nature communications readers should benefit from an analysis that is easily accessible to the researchers and that is straightforward to perform and that would help in understanding the results.

5) There are interpretation mistakes in the presentation of MR results. In a rewrite of this piece, the authors would do well to include an MR statistical expert on their team to guide the interpretation of their results.

l. 511-512

...used to evaluate if CP/CPPS causally affects BPH or CaP since these phenotypes often appear together

In this first sentence to introduce the MR results, the authors mention one direction of association: CP/CPPS causally affecting BPH. However, the results actually show more evidence for the opposite direction.

Specifically:

The MR results in the paragraph starting line 516 contain the following statements:

1. “Mixed results” line 516
2. “consistent inverse associations” line 523
3. “MR Egger did not show a significant association.” L. 521.

Looking at results in ST9a that these comments refer to, I am seeing a negative z for all methods presented except Egger regression. Also, three out of five methods are nominally significant if we consider a p of 0.05 as the significance cut-off.

Egger regression is not significant. Given that Egger supports absence of pleiotropy, and has lower power, we can still consider the three out of five methods tested to show statistical significance as showing evidence of causality. Thus I agree with statement 1 and 3, but statement 2 is not true since it would imply all five methods give a negative z.

L. 524-525:

The authors suggest "The findings indicate that BPH may be causative of CP/CPPS." The evidence for this statement is Table S9a.

This is a misunderstanding by the authors. The findings in Table S9a indicate a causal relationship where CP/CPPS might reduce the risk of BPH. This is very different from BPH causing CP/CPPS. In other words, a reduction in risk in one direction examined by MR does not imply an opposite direction for the causal arrow like the authors assert.

Evidence from Table S9b does seem to suggest that BPH may indeed be causative of CP/CPPS (with BPH potentially reducing the risk of CP/CPPS). All MR methods are statistically significant, and the direction of association is negative. However, the reduction of risk is not mentioned by the authors.

The authors summarize these results as reflecting a "complex interplay." Unfortunately, these results are difficult to interpret. The strongest evidence supports BPH as protective against CP/CPPS – this does not seem intuitive. How would a pathway that seems to decrease risk of CP-CPPS but increase risk of BPH be helpful in devising a therapeutic strategy? In the abstract, the authors suggest that these MR results "could inform therapeutic strategies."

Reviewer #2

(Remarks to the Author)

Reviewer 2: All comments have been Addressed

Reviewer 3:

REVIEWER COMMENT: Rosenthal S.B. et al performed the first GWAS of Chronic Prostatitis/Chronic Pelvic Pain Syndrome (CP/CPPS) in men with 14,575 cases and 568,820 controls without CP/CPPS. They identified a total of six loci across ancestry groups, including a Hispanic ethnicity specific locus. The majority of these loci have been associated with related phenotypes including benign prostatic hyperplasia (BPH) and prostate-specific antigen (PSA), supporting the hypothesis of the multifaceted etiology of CP/CPPS. They follow the GWAS with appropriate methods to further identify pleiotropy between CP/CPPS and other related phenotypes (CaP and BPH). They further demonstrate an interconnection between CP/CPPS and neurological disorders. The work presented supports the knowledge of the field and provides routes of further investigation into the pleiotropy of CP/CPPS and BPH. Overall, the authors performed the study well, they utilized appropriate methods, and the results are novel and clearly communicated.

AUTHOR RESPONSE: We thank the reviewer for their comments.

1) REVIEWER COMMENT: 1. It would be beneficial to include the X chromosome in the analyses.

AUTHOR RESPONSE: We agree with the reviewer that including X-chromosome data would be valuable, but it was not available at the time of analysis. We hope that this will be possible in future.

REVIEWER RESPONSE: X chromosome data is available in MVP.

2) REVIEWER COMMENT: To further the authors' point of the shared genetic architecture between CP/CPPS and other diseases, it could be beneficial to conduct a PheWAS of the identified loci (potentially just the clumped and pruned variants utilized in the MR) to investigate the other associated phenotypes.

AUTHOR RESPONSE: As the reviewer suggests, we have now included a PheWAS of the 6 independent significant SNPs from the 3 GWS loci and included any other traits that were significant from the GWAS catalog as a supplemental table (Supplemental Table 6).

REVIEWER RESPONSE: Based on written results and Supplemental Table 6, it would be more apt to remove the term PheWAS and continue with GWAS catalog trait lookup.

3) REVIEWER COMMENT: While age was not included in the model, it would be worthwhile to conduct a sensitivity analysis including age as a covariate.

AUTHOR RESPONSE: Conducting the analyses with age at MVP enrollment as a covariate yielded the same three genome-wide significant loci and lead SNPs as our original analysis. The genetic correlation between the EUR CP/CPSS GWAS without age and the EUR CP/CPSS GWAS with age was 0.9922 (SE=0.181). We report these findings in the Results section.

REVIEWER RESPONSE: While I appreciate the sensitivity analysis; I am curious why they used age and MVP enrollment and not age at first CP/CPSS diagnosis.

4) REVIEWER COMMENT: As the authors use the CC-GWAS method for CaP and BPH, it would be worthwhile to include the MR intercepts as further support of potential pleiotropy between the phenotypes.

AUTHOR RESPONSE: We have added the MR Egger intercepts (all non-significant) to the results section.

REVIEWER RESPONSE: I am satisfied with the inclusion of the MR Egger intercepts as well as the language added in the manuscript.

5) REVIEWER COMMENT: While informative, Figure 1B is difficult to read at the presented font size.

AUTHOR RESPONSE: We have increased the font size in Figure 1B to improve readability.

REVIEWER RESPONSE: I am satisfied with the updated title.

6) REVIEWER COMMENT: Line 432: Authors state 1,466 traits while the methods state 823 traits.

AUTHOR RESPONSE: We have clarified in the methods that phenotypes were restricted to those with sufficient heritability with the statement: "To ensure robust findings, we restricted our analyses from the 1437 available phenotypes to those displaying a SNP heritability z-score greater than 4, yielding 823 phenotypes for evaluation."

REVIEWER RESPONSE: I am satisfied with the edited wording.

7) REVIEWER COMMENT: Line 455 – 458: If the authors could clarify this section. Based on the authors description of the exposure (CP/CPSS) and outcome (BPH) and result, it would seem that CP/CPSS may be causative of BPH, however, it is written as BPH may be causative of CP/CPSS.

AUTHOR RESPONSE: We appreciate the reviewer's request for clarification. We have clarified the following points in our methods, results, and discussion. We explored the causal relationships between CP/CPSS and BPH using bidirectional Mendelian randomization (MR). Our primary analysis modeled CP/CPSS as the exposure and BPH as the outcome, revealing a significant inverse relationship in several MR methods. These findings suggest that CP/CPSS may indeed be causative of BPH.

However, to fully explore the relationship, we also conducted a reverse MR analysis, modeling BPH as the exposure and CP/CPSS as the outcome. This analysis yielded significant results, suggesting that BPH may contribute to the development of CP/CPSS. These bidirectional results highlight a complex interplay between CP/CPSS and BPH and suggest that therapeutic interventions targeting BPH might alleviate symptoms of CP/CPSS, and conversely, managing CP/CPSS might influence the risk of BPH. We acknowledge that this bidirectional relationship might seem confusing and have clarified it in the manuscript to ensure that both directions of causality are communicated. We have revised the relevant sections to emphasize that while CP/CPSS appears to reduce the risk of BPH based on the initial analysis, the reverse relationship also holds true based on our additional findings.

As we describe in our methods and results, we have also included CASUE analyses that employ full GWAS summary statistics to test MR models of causality. Our results are consistent across methods.

REVIEWER RESPONSE: I would recommend adding both the term 'bidirectional' as well as a sentence on using BPH as the exposure and CP/CPSS as the outcome into the mendelian randomization section of the Methods. Otherwise, I am happy with the clarification.

8) REVIEWER COMMENT: While the title includes prostatitis, suggesting a study of men, it should include the denotation of including only men in the study following the Nature Reporting Summary.

AUTHOR RESPONSE: The title now reads: "The Genetic Basis of Chronic Prostatitis/Chronic Pelvic Pain Syndrome: Insights from a Large-Scale Multi-ancestry Study of Men."

REVIEWER RESPONSE: I am satisfied with the updated title.

Reviewer #3

(Remarks to the Author)

Version 2:

Reviewer comments:

Reviewer #1

(Remarks to the Author)

Abstract:

We identified the genetic correlations...

Word choice problem: the genetic correlations were estimated. Identified gives the impression they have a fixed value and that the study was able to find those values. That is misleading.

Reviewer 1 responses:

Responses to points 1 and 2 seem OK.

New text on p. 8:

Did all the authors read this?

SNPs on high LD ($r^2 \geq 0.001$) with a more significant SNP with a 1Mb window would be pruned.

On? Would – under what circumstances?

New text results first line: lifetime phenotypic prevalence

This seems to imply the veterans were all followed until death. Is this true? If not, then the wording used here is quite confusing.

Reviewer comment 5:

Results section: "Among Veterans with CP/CPSP, 76.26% had BPH"

Given that over 75% percent of those with CP/CPSP are the same individuals as those with BPH, it seems it is not surprising that they share causal variants (Figure 2)? It is not surprising that genetic correlation is high also. While the results section has been updated to present data, this new data has not been used to provide interpretation in relation to this new finding.

Discussion:

The strong genetic correlation between CP/CPSP and BPH, and the GWAS catalog associations, suggest that the genetic causes may be very similar for these conditions.

Isn't high sample overlap a more straightforward explanation for the high genetic correlation?

Also... "suggesting that these traits are deeply linked."

Many who have CP/CPSP also have BPH.

New text; p.9: MVP analyses showed lifetime phenotypic prevalence of CP/CPSP at 2.23% (22,290/962,151), CaP at 3.30% (31,715/962,151), and BPH at 2.32% (22,290/962,151).

There is an issue since the percent and count of CP/CPSP and BPH are identical. The expectation is that the number of individuals with BPH would be higher than the number with CP/CPSP.

An error present in a previous version of the manuscript seems like it was not corrected on p.11:

GWAS of Chronic Prostatitis/Chronic Pelvic Pain Syndrome had a regulomeDB score of 3a or higher, indicating possible regulatory relationships. (Table S3).

lower regulomeDB scores are indicative of regulatory relationships.

Figure 1 legend: I don't see an indication of what S1, S2 and S3 mean. It seems M1-25 are the genes but this is not explicit

in the legend.

The phrase "collapsing the GWAS signal within the gene set of interest" is not the clearest way to describe what MAGMA does.

Better: to draw from phrases such as: aggregating GWAS signals across all SNPs within each gene; MAGMA incorporates all genes regardless of their statistical significance

Under Relationship to BPH and CaP

Better not to use the term: causal interplay. This is vague and suggests some kind of network of causal relationships. Instrumental variables are strictly defined. Each causal arrow is evaluated explicitly. The authors should say what arrows were tested, and then what arrows have associated causal evidence.

New text page 16:

"For CP/CPPS against BPH, ... suggesting a causal relationship between CP/CPPS and BPH". Similarly, For BPH against CP/CPPS, the causal model was a better fit than the null ($p=.01$) and sharing models ($p=.03$), suggesting a causal relationship between CP/CPPS and BPH".

Here opposite directions of causality are tested (green), and the conclusion by the authors is the same (blue). I am not seeing an interpretation that considers the meaning of the directionality.

Reviewer 3

1. It seems the authors could have performed a GWA scan of the X chromosome among men and provided a Manhattan plot with those findings? The X chromosome test statistics could simply be appended to those for the autosomes. It is understandable that downstream analyses would not be conducted for now on X chromosome data, but the Manhattan plot would at least provide basic findings.

Reviewer #2

(Remarks to the Author)

Reviewer #3

(Remarks to the Author)

Reviewer #4

(Remarks to the Author)

I was invited to review this manuscript specifically to comment on the MR analyses. It appears that there was an error in previously submitted analyses, but this has now been resolved. Overall, the MR analyses are performed somewhat algorithmically and uncritically, but also clearly. When I see bidirectional results from an MR study, I think this is likely to represent shared aetiology (i.e. common causes of both traits) rather than literal causal effects in both directions. However, this is a matter of interpretation. Overall, the MR analyses are not a barrier to the publication of this manuscript.

As for the rest of the work, these sort of untargeted "throw everything at the wall" analyses aren't my cup of tea if I'm being honest, but I'm sure some people like them. But again, that is a matter of preference.

Minor comment:

"HARE ensures accurate classification using GIA..." - I doubt any statistical method ensures anything - 'ensure' seems too strong here.

Version 3:

Reviewer comments:

Reviewer #1

(Remarks to the Author)

The authors have provided adequate responses to the points raised. However, some of the issues, such as the incorrect use

of verb tenses or the data reporting error noted under point 5, suggest a need for greater attention to detail. These types of errors may occur elsewhere in the manuscript and ideally the authors should conduct a more comprehensive check to ensure the overall consistency and accuracy of their work.

REVIEWER 1

1) REVIEWER COMMENT: This study addresses an important biomedical question, specifically the common genetics between Chronic Prostatitis/Chronic Pelvic Pain Syndrome and other conditions affecting the prostate and other conditions generally. The study is based on the Million Veteran Program study including 590,000 veterans of European, African and Hispanic ancestry from the United States, of which 14,575 have CP/CPPS. The most interesting finding are lines of evidence supporting common genetics between CP/CPPS and benign prostate hyperplasia but not between CP/CPPS and prostate cancer. Overlapping genetics between CP/CPPS and other traits are also examined highlighting neurological function, matching some well-known findings of overlapping genetics across pain phenotypes and neurological function.

While the overall idea of the study is one that merits strong consideration in terms of a prominent venue for publication, the manuscript in its present form suffers from poor or incomplete execution in several key ways that weaken the whole to the point where a thorough re-working of the manuscript would be necessary. These impact on how convincing or well-supported the conclusions of the study are.

AUTHOR RESPONSE: We thank the reviewer for their comments. We address specific requests and clarify our approach in the response below.

2) REVIEWER COMMENT: The numbers of study subjects classified as cases who were attributed an ICD code of one or the other out of CP and/or CPPS phenotypes is not detailed. How many study subjects were attributed codes for both phenotypes? Based on information provided in the first para of the introduction, most who have CP also have CPPS type symptoms. However, what is not detailed is the proportion of individuals with CPPS who have CP. Both sides would need to be examined carefully in a study that now puts the two together presenting these as a single phenotype, and in two ways – via the literature, and via the dataset being used.

AUTHOR RESPONSE: Chronic prostatitis/chronic pelvic pain syndrome (CP/CPPS) is recognized as a single condition despite the combined name potentially causing some confusion. This terminology is widely accepted in the medical community to describe a complex syndrome characterized by chronic pelvic pain. Consequently, CP/CPPS represents a single phenotype encompassing a range of symptoms and clinical presentations. To ensure clarity and avoid misunderstanding, we have explicitly detailed this classification in the introduction, paragraph one, which reads: "Chronic Prostatitis/Chronic Pelvic Pain Syndrome (CP/CPPS) is a prevalent and debilitating condition affecting 2-10% of men. Over 90% of all prostatitis diagnoses are CP/CPPS, making it the most prevalent prostatitis diagnosis. Symptoms include persistent pelvic pain and discomfort, sexual dysfunction, and urinary symptoms for at least three months, which can severely impair quality of life. "

3) REVIEWER COMMENT: Basic univariate analyses of the phenotypes considered in the study should be performed.

E.g. the reader should be presented with the proportion of CPPS individuals who have benign prostate hyperplasia. What about those who have prostate cancer? What about the distribution of the 12 phenotypes according to CP and CPPS status?

It is impossible to tell from the information provided whether the genetic overlap with phenotypes like backpain may be due to a higher proportion of CPPS compared to CP in the combined phenotype.

AUTHOR RESPONSE: The current study utilized genome-wide association study (GWAS) summary statistics from different studies and samples, as detailed in Supplemental Table 7. Consequently, we cannot provide the phenotypic overlap between various conditions (e.g., chronic back pain in the UK Biobank and CP/CPPS in the MVP cohort). Using summary statistics from various studies is a common practice in conducting post-GWAS analyses. We agree that future research should expand these analyses to classify phenotypic overlaps more comprehensively and within the same sample.

The genetic correlations with 12 other conditions are based on logistic regression results from GWAS. As detailed in our methods section, LD Score Regression (LDSC) evaluates genetic signals across different samples and prevalence rates, estimating the degree of genetic overlap. This methodology allows us to infer genetic correlations even when direct phenotypic comparisons are impossible.

4) REVIEWER COMMENT: The second introduction paragraph evokes “unpredictable symptomatology” – what is unpredictable: the order of symptoms over time, the magnitude of importance of each element; the number of possible symptoms that must be present for a diagnosis to be positive? Are the authors saying that the diagnosis process for identifying this condition is highly problematic and therefore it isn’t clear what CP and or CPPS is anyway? How is the diagnosis made in the clinic? Would there be a lot of disagreement across physicians on attributing a CP or CPPS diagnosis? The starting point of the present study is so fuzzy that none of the results can be interpreted in any meaningful way, it seems.

AUTHOR RESPONSE: We have removed the word “unpredictable.” There is only one condition called CP/CPPS. To enhance clarity and improve the interpretability of our results, we have revised the introduction to explicitly define CP/CPPS, describe its symptomatology, and include this in limitations for future studies to address. The introduction now reads: “CP/CPPS is characterized by heterogeneity and transient symptomatology that can fluctuate over time in intensity and order of appearance.”

For a diagnosis of CP/CPPS, clinicians rely on patient history, symptom presentation, and exclusion of other conditions. Common symptoms include chronic pelvic pain, urinary symptoms, and sexual dysfunction that persist for at least three months. While some variability in diagnoses among physicians is possible due to the subjective nature of symptom reporting, clinical guidelines and standardized questionnaires used in clinical practice provide a more consistent framework for diagnosis.

5) REVIEWER COMMENT: Basic reporting guidelines were not followed. Examination of the IV assumptions was not undertaken. 5 methods were tried and one reported in the results section. Going to the table S8, only two methods give a nominally statistically significant result. The robust Egger method is not statistically significant. There is no process for screening the instrumental variables. What were the instrumental variables used? What about sensitivity analyses? Logic in the paragraph II. 453-465 does not hold up: the sentence “we modeled...” suggests a uni-directional approach was used. Later it is suggested a bi-directional approach was used. This paragraph leaves an impression of confusion. These results simply cannot be interpreted or trusted.

AUTHOR RESPONSE: Thank you for your detailed feedback. We have addressed the concerns raised regarding the examination of MR assumptions. We report that we calculated the F-statistics for each SNP and ensured that all selected SNPs had F-statistics greater than 10, confirming the strength of the instruments and the variance explained by each SNP. We screened the SNPs for associations with potential confounders. Furthermore, we report the MR Egger intercepts. Additionally, we performed sensitivity analyses, including MR-Egger regression, MR-PRESSO, leave-one-out analysis, and Cochran's Q test, to detect and correct for potential pleiotropy. Finally, we repeated our MR analyses using CASUE, designed to use full GWAS summary statistics to overcome the limitations inherent in traditional MR approaches described above. Our findings are consistent across methods.

We have revised the methods section to outline our methodology and sensitivity analyses conducted. We have provided a comprehensive analysis of the results and clarified the logic in interpreting the findings. We believe these revisions address the reviewer's concerns and provide a clearer description of our analyses. Our results or conclusions did not change.

6) REVIEWER COMMENT: Paragraph II. 610-623 goes into lots of detail about a well know truism relating to common complex disease: twin and family-based study estimates of heritability are always higher than SNP chip heritability. This paragraph adds nothing that is specific to prostate conditions.

AUTHOR RESPONSE: Thank you for your comment. We appreciate the perspective and agree that the discrepancy between twin/family-based heritability estimates and SNP-based heritability is a well-known phenomenon in studying complex diseases and among the broader genetic community. However, we believe this paragraph is crucial for several reasons:

- Specificity to CP/CPPS: While the general concept of "missing heritability" is well-known, this study provides the first comparative heritability estimates specifically for CP/CPPS, a condition that has not been as extensively studied in this context. Our findings offer a unique contribution by incorporating the only available estimates for this condition, thus filling a significant gap in the literature. We believe it would be an oversight not to compare our heritability estimates to those of previous twin study results, and doing so requires an explanation of the differences in methods.
- Context for Broader Readership: Nature Communications has a broad readership, including many who might not be intimately familiar with the nuances of heritability estimates and their implications in CP/CPPS research. We believe including this discussion helps contextualize our findings and underscores the importance of diverse methodological approaches in understanding the genetic underpinnings of CP/CPPS.
- Highlighting Methodological Differences: The paragraph discusses the discrepancy and its reasons, emphasizing the importance of considering different genetic factors and methodological approaches. This is particularly relevant for advancing the field and guiding future research directions.

7) REVIEWER COMMENT: Lines 698 etc. Are there no other biobanks that have recorded prostate condition data? If the authors tried to go to other biobanks and searched for other resources with both prostate condition data and genotyping data but the data was not available, this should be detailed explicitly. What needs to be done to remedy this situation so that prostate conditions get adequate representation from biobanks?

AUTHOR RESPONSE: Thank you for this astute question. After a careful review of publicly available summary statistics for CP/PPS, we found no such results. We agree with the reviewer that other large biobanks should pay greater attention to CP/PPS, and better representation should be encouraged. We hope our results will inspire further studies in other samples and contribute to a better understanding of this condition. Our limitations section reads: "The absence of replication datasets within other biobanks, including UKB and All of US, highlights the need for collaborative efforts and the collection of additional CP/PPS data."

8) REVIEWER COMMENT: Table 1: Why aren't tests presented showing differences in case proportions by ancestry group? Why aren't percentages of the different ancestries as a part of the whole MVP presented?

AUTHOR RESPONSE: The proportion of cases and controls have been added to Table 1 in parentheses. The percentage of HARE ancestry/race groups is now described in the first paragraph of the results, that reads:

"The sample included 583,395 participants with 14,575 CP/PPS cases and 568,820 controls stratified by three HARE race/ancestry groups, with EUR comprising 73.76% of the sample, followed by AFR (18.18%), and HIS (8.06%). Table 1 shows the average age and CP/PPS prevalence by HARE groups. CP/PPS prevalence for the three groups was 2.50% and varied across HARE groups, with 2.33% of EUR (10,035 of 430,306 individuals), 3.35% of AFR (3,553 of 106,081 individuals), and 2.10% of HIS (987 of 47,008 individuals). Average age varied across the HARE groups from 64.64 (SD = 13.35) years for EUR to 55.84 (15.63) years for HIS and was higher for CP/PPS cases than controls in all ancestries ($p < 0.001$). Prevalence rate did not differ between European vs Hispanic ancestry ($p = .002$)."

A footnote for Table 1 indicates that the "Prevalence rate was significantly different ($p < .001$) for all HARE ancestry/race groups contrasts except EUR vs. HIS ($p = .002$)."

9) REVIEWER COMMENT: Line 356: the authors seem to be confused on how to interpret RegulomeDB scores. Lower scores show greatest evidence of regulatory involvement.

AUTHOR RESPONSE: We have clarified our wording in the text: "Of 118 SNPs significantly associated with CP/PPS in the three GWS loci ($p < 5E-8$), 26 had a RegulomeDB score of 3a or lower, indicating possible regulatory relationships." Our results remain the same.

10) REVIEWER COMMENT: Gene symbols are sometimes in italics, sometimes not. Check your manuscript before submitting it.

AUTHOR RESPONSE: GENE symbols have been italicized.

REVIEWER 2

1) REVIEWER COMMENT: In the GWAS Cohort and Phenotype section, the term "ancestry" is used. Based on the context, I believe the authors are likely referring to the categorical HARE race variable for MVP participants. I think that moving the Race, Ethnicity, and Ancestry section

before the GWAS Cohort and Phenotype section, and stating that the groupings of dropped individuals were based on the HARE race/ancestry categories would perhaps avoid misunderstandings with readers who are not familiar with the HARE approach.

In subsequent mentions elsewhere in the paper, I also think it would be prudent to adjust the terminology to specify that you are using the HARE race/ancestry categories to stratify participants into analysis groups.

This is clearly presented in Table 1, but I feel the precise language should be used throughout the paper. For example, then sentence: "The CP/CPPS GWAS included 10,035 cases and 420,271 controls of European ancestry." On line 343, would be more accurately stated: "The CP/CPPS GWAS included 10,035 cases and 420,271 controls from the HARE race/ancestry category." This is because those individuals are predominantly European and self-identify as "non-Hispanic white" but are not necessarily entirely European.

AUTHOR RESPONSE: Thank you for this suggestion; it greatly streamlines the methods. We have moved the Race, Ethnicity, and Ancestry section before the GWAS Cohort and Phenotype section and defined the ancestries (i.e., EUR, HIS, AFR) in the Race, Ethnicity, and Ancestry section. It now reads:

"Here, we use the term "Hispanic" (HIS) for the HARE race and ethnicity groups comprised of individuals who are Latino or Hispanic, the term "European" (EUR) for individuals who are White but not Hispanic, and "African" (AFR) for individuals who are Black but not Hispanic. People of East Asian and South Asian ancestry were not analyzed due to the low numbers in MVP. Hereafter, we refer to the HARE groups used in analyses as HIS, EUR, and AFR."

We have also adjusted the language in the first paragraph of the results to reflect the use of HARE-based categories. It now reads:

"The sample included 583,395 participants with 14,575 CP/CPPS cases and 568,820 controls stratified by three HARE race/ancestry groups, with EUR comprising 73.76% of the sample, followed by AFR (18.18%), and HIS (8.06%). Table 1 shows the average age and CP/CPPS prevalence by HARE groups. CP/CPPS prevalence for the three groups was 2.50% and varied across HARE groups, with 2.33% of EUR (10,035 of 430,306 individuals), 3.35% of AFR (3,553 of 106,081 individuals), and 2.10% of HIS (987 of 47,008 individuals). Average age varied across the HARE groups from 64.64 (SD = 13.35) years for EUR to 55.84 (15.63) years for HIS and was higher for CP/CPPS cases than controls in all ancestries ($p < 0.001$). Prevalence rate did not differ between European vs Hispanic ancestry ($p = .002$)."

2) REVIEWER COMMENT: The abbreviations EUR, AFR, HIS are established on line 205 without defining them. I think it would be fine to state that these abbreviations refer to the HARE race/ancestry categories and use them on subsequent mentions. In general I see many instances of switching between using these abbreviations and XX ancestry. I think it is important to clean this up and use a consistent naming convention based on the approach.

AUTHOR RESPONSE: Please see the previous response. We have defined the three groups in the methods and cleaned up the naming across the manuscript to be consistent with the three HARE race/ancestry categories.

3) REVIEWER COMMENT: The IGC should be changed to the Greek lambda (line 394)

AUTHOR RESPONSE: Changed to Greek lambda.

4) REVIEWER COMMENT: Were all the statistics presented in the SNP-based heritability section derived from analysis of EUR data? Perhaps the HIS was too small, but perhaps the AFR data could provide an estimate.

AUTHOR RESPONSE: We clarified in the SNP-based heritability section that the results are based on EUR.

We clarify our limitations (p.21): " First, although this was the first CP/CPPS GWAS study to include multiple ancestry groups, people of East Asian and South Asian ancestry groups within the MVP cohort were small and, therefore, not analyzed. Heritability estimates were conducted on EUR sample where a sufficiently large sample size was available to produce reliable estimates."

5) REVIEWER COMMENT: Fst is usually presented with a subscript

AUTHOR RESPONSE: Modified with subscript.

6) REVIEWER COMMENT: GTEx should be spelled out and cited on first mention

AUTHOR RESPONSE: Added to the methods, p7.

7) REVIEWER COMMENT: Is there any measure of how well a single instance of an ICD code classifies either of these outcomes (i.e. a PPV compared to expert review of records)? If not, this should be acknowledged as a limitation and an opportunity for improvement in future studies in the Discussion.

AUTHOR RESPONSE: Thank you for your valuable comment. We acknowledge the importance of validating ICD code-based classifications to ensure the accuracy of our findings. Unfortunately, in this study, we did not have the opportunity to validate through chart review of records. We agree that this is an important aspect that should be acknowledged as a limitation and considered for improvement in future studies. The limitation section now reads:

"Assessment of CP/CPPS was primarily based on EHR and ICD codes. Such data are valuable for large-scale studies. However, EHR-based assessment may not capture the full spectrum of clinical manifestations and may be subject to inconsistencies in coding and documentation. Because CP/CPPS is a complex and heterogeneous condition with various clinical presentations, it is unlikely that all cases of CP/CPPS to be adequately reflected in ICD codes and we were not able to validate the ICD codes using chart review, a imitation that can be overcome by future studies. EHR may also be incomplete because patients may seek care elsewhere or have not yet developed the conditions. Future efforts should employ more comprehensive clinical assessment and validation studies to enhance the precision of genetic studies of CP/CPPS. "

REVIEWER 3

1) REVIEWER COMMENT: Rosenthal S.B. et al performed the first GWAS of Chronic Prostatitis/Chronic Pelvic Pain Syndrome (CP/CPPS) in men with 14,575 cases and 568,820 controls without CP/CPPS. They identified a total of six loci across ancestry groups, including a Hispanic ethnicity specific locus. The majority of these loci have been associated with related phenotypes including benign prostatic hyperplasia (BPH) and prostate-specific antigen (PSA), supporting the hypothesis of the multifaceted etiology of CP/CPPS. They follow the GWAS with appropriate methods to further identify pleiotropy between CP/CPPS and other related phenotypes (CaP and BPH). They further demonstrate an interconnection between CP/CPPS and neurological disorders. The work presented supports the knowledge of the field and provides routes of further investigation into the pleiotropy of CP/CPPS and BPH. Overall, the authors performed the study well, they utilized appropriate methods, and the results are novel and clearly communicated.

AUTHOR RESPONSE: We thank the reviewer for their comments.

2) REVIEWER COMMENT: 1. It would be beneficial to include the X chromosome in the analyses.

AUTHOR RESPONSE: We agree with the reviewer that including X-chromosome data would be valuable, but it was not available at the time of analysis. We hope that this will be possible in future.

3) REVIEWER COMMENT: To further the authors' point of the shared genetic architecture between CP/CPPS and other diseases, it could be beneficial to conduct a PheWAS of the identified loci (potentially just the clumped and pruned variants utilized in the MR) to investigate the other associated phenotypes.

AUTHOR RESPONSE: As the reviewer suggests, we have now included a PheWAS of the 6 independent significant SNPs from the 3 GWS loci and included any other traits that were significant from the GWAS catalog as a supplemental table (Supplemental Table 6).

4) REVIEWER COMMENT: While age was not included in the model, it would be worthwhile to conduct a sensitivity analysis including age as a covariate.

AUTHOR RESPONSE: Conducting the analyses with age at MVP enrollment as a covariate yielded the same three genome-wide significant loci and lead SNPs as our original analysis. The genetic correlation between the EUR CP/CPPS GWAS without age and the EUR CP/CPPS GWAS with age was 0.9922 (SE=0.181). We report these findings in the Results section.

5) REVIEWER COMMENT: As the authors use the CC-GWAS method for CaP and BPH, it would be worthwhile to include the MR intercepts as further support of potential pleiotropy between the phenotypes.

AUTHOR RESPONSE: We have added the MR Egger intercepts (all non-significant) to the results section.

6) REVIEWER COMMENT: While informative, Figure 1B is difficult to read at the presented font size.

AUTHOR RESPONSE: We have increased the font size in Figure 1B to improve readability.

7) REVIEWER COMMENT: Line 432: Authors state 1,466 traits while the methods state 823 traits.

AUTHOR RESPONSE: We have clarified in the methods that phenotypes were restricted to those with sufficient heritability with the statement: "To ensure robust findings, we restricted our analyses from the 1437 available phenotypes to those displaying a SNP heritability z-score greater than 4, yielding 823 phenotypes for evaluation. "

8) REVIEWER COMMENT: Line 455 – 458: If the authors could clarify this section. Based on the authors description of the exposure (CP/CPPS) and outcome (BPH) and result, it would seem that CP/CPPS may be causative of BPH, however, it is written as BPH may be causative of CP/CPPS.

AUTHOR RESPONSE: We appreciate the reviewer's request for clarification. We have clarified the following points in our methods, results, and discussion. We explored the causal relationships between CP/CPPS and BPH using bidirectional Mendelian randomization (MR). Our primary analysis modeled CP/CPPS as the exposure and BPH as the outcome, revealing a significant inverse relationship in several MR methods. These findings suggest that CP/CPPS may indeed be causative of BPH.

However, to fully explore the relationship, we also conducted a reverse MR analysis, modeling BPH as the exposure and CP/CPPS as the outcome. This analysis yielded significant results, suggesting that BPH may contribute to the development of CP/CPPS. These bidirectional results highlight a complex interplay between CP/CPPS and BPH and suggest that therapeutic interventions targeting BPH might alleviate symptoms of CP/CPPS, and conversely, managing CP/CPPS might influence the risk of BPH. We acknowledge that this bidirectional relationship might seem confusing and have clarified it in the manuscript to ensure that both directions of causality are communicated. We have revised the relevant sections to emphasize that while CP/CPPS appears to reduce the risk of BPH based on the initial analysis, the reverse relationship also holds true based on our additional findings.

As we describe in our methods and results, we have also included CASUE analyses that employ full GWAS summary statistics to test MR models of causality. Our results were consistent across methods.

9) REVIEWER COMMENT: While the title includes prostatitis, suggesting a study of men, it should include the denotation of including only men in the study following the Nature Reporting Summary.

AUTHOR RESPONSE: The title now reads: "The Genetic Basis of Chronic Prostatitis/Chronic Pelvic Pain Syndrome: Insights from a Large-Scale Multi-ancestry Study of Men."

Reviewer 1

- 1) **REVIEWER COMMENT:** MiXer is mentioned as a jargony black box rather than in language that can be interpreted by a scientist who has not used the software, especially for the general audience targeted by Nature Communications. The authors should interpret what MiXer contributed without using the acronym MiXer.

AUTHOR RESPONSE: We replaced the abstract phrasing. It now reads "Results of bivariate causal mixture modeling indicate that some of the same genetic variants likely contribute to the development of CP/CPPS, BPH, and CaP. "

- 2) **REVIEWER COMMENT:** Introduction Some clarity is still missing from the explanation of the CP/CPPS diagnosis. A sentence like "CP/CPPS is a diagnostic label attributed to chronic prostatitis cases for which a concomitant infection is not detected" is needed here as the provided text does not explain that a diagnostic algorithm is followed by physicians with this label being one possible classification.

AUTHOR RESPONSE: We have clarified the phrasing. The introduction now reads: "CP/CPPS diagnosis relies on the reported symptoms of prostatitis, occurring in the absence of detectable urinary tract infections and other identifiable disorders. This diagnostic approach serves to differentiate CP/CPPS from acute bacterial prostatitis, chronic bacterial prostatitis, and asymptomatic inflammatory prostatitis. "

- 3) **REVIEWER COMMENT:** "In the multi-ancestry meta-analysis, we identify six novel loci associated with CP/CPPS 158 risk, an increase from three significant genome-wide loci found in the European cohort."

à Are you saying the three loci identified among Europeans are not novel? Does this mean they have been reported elsewhere? Or do you mean there are three that are also found in other ethnicities, showing validation across different ancestries? If they are not novel and have been reported elsewhere, references should be provided.

AUTHOR RESPONSE: The introduction, last paragraph now reads: "In the multi-ancestry meta-analysis, we identify three additional novel loci associated with CP/CPPS risk, in addition to the three novel significant genome-wide loci found in the European cohort alone."

- 4) **REVIEWER COMMENT:** Under reviewer 1, comment 1, the authors did not take the comment into consideration.

AUTHOR RESPONSE: We believe this comment refers to the request below.

- 5) **REVIEWER COMMENT:** A basic statistical analysis of the phenotypes available to researchers in the MVP cohort should be provided. The authors have chosen to ignore this reviewer comment.

The interpretation of genetic correlations is much more meaningful if this can be done in the light of phenotypic correlations. Nature communications readers should benefit from an analysis that is easily accessible to the researchers and that is straightforward to perform and that would help in understanding the results.

AUTHOR RESPONSE: We apologize for the misunderstanding. The first paragraph of the results section now reads:

"MVP analyses showed lifetime phenotypic prevalence of CP/CPPS at 2.23% (22,290/962,151), CaP at 3.30% (31,715/962,151), and BPH at 2.32% (22,290/962,151). Among Veterans with CP/CPPS, 76.26% had BPH, and 7.59% had CaP. Of the individuals with BPH, 5.5% had CP/CPPS, and 4.2% had CaP. In CaP cases, 5.3% had CP/CPPS, and 40.6% had BPH. Tetrachoric correlations showed positive associations, strongest between CP/CPPS and BPH ($r_{tet} = 0.47$ [95% CI = 0.47 – 0.47, $p < .001$]), followed by CP/CPPS and CaP ($r_{tet} = 0.18$ [95% CI = 0.18 – 0.18, $p < .001$]), and CaP and BPH ($r_{tet} = 0.11$ [95% CI = 0.10 – 0.11, $p < .001$])."

- 6) **REVIEWER COMMENT:** There are interpretation mistakes in the presentation of MR results. In a rewrite of this piece, the authors would do well to include an MR statistical expert on their team to guide the interpretation of their results.

I. 511-512

...used to evaluate if CP/CPPS causally affects BPH or CaP since these phenotypes often appear together

In this first sentence to introduce the MR results, the authors mention one direction of association: CP/CPPS causally affecting BPH. However, the results actually show more evidence for the opposite direction.

Specifically: The MR results in the paragraph starting line 516 contain the following statements:

1. "Mixed results" line 516

2. "consistent inverse associations" line 523

3. "MR Egger did not show a significant association." L. 521.

Looking at results in ST9a that these comments refer to, I am seeing a negative z for all methods presented except Egger regression. Also, three out of five methods are nominally significant if we consider a p of 0.05 as the significance cut-off. Egger regression is not significant. Given that Egger supports absence of pleiotropy, and has lower power, we can still consider the three out of five methods tested to show statistical significance as showing evidence of causality. Thus I agree with statement 1 and 3, but statement 2 is not true since it would imply all five methods give a negative z.

L. 524-525:

The authors suggest "The findings indicate that BPH may be causative of CP/CPPS." The evidence for this statement is Table S9a.

This is a misunderstanding by the authors. The findings in Table S9a indicate a causal relationship where CP/CPPS might reduce the risk of BPH. This is very different from BPH causing CP/CPPS. In other words, a reduction in risk in one direction examined by MR does not imply an opposite direction for the causal arrow like the authors assert.

Evidence from Table S9b does seem to suggest that BPH may indeed be causative of CP/CPPS (with BPH potentially reducing the risk of CP/CPPS). All MR methods are statistically significant, and the direction of association is negative. However, the reduction of risk is not mentioned by the authors.

The authors summarize these results as reflecting a "complex interplay." Unfortunately, these results are difficult to interpret. The strongest evidence supports BPH as protective against CP/CPPS – this does not seem intuitive. How would a pathway that seems to decrease risk of CP-CPPS but increase risk of BPH be helpful in devising a therapeutic strategy? In the abstract, the authors suggest that these MR results "could inform therapeutic strategies."

AUTHOR RESPONSE: We thank the reviewer for the careful review and bringing this issue

to our attention. We have identified a mistake in our analysis, which resulted in the direction of effect being reversed between CP/CPPS and BPH in the results from MR-Base (the issue was not present in the MR results using the CAUSE software). The issue we identified was that the effect allele and non-effect allele were miscoded for BPH, relative to CP/CPPS meaning that the direction of effect we reported was opposite to the true effect. We apologize for this mistake, and have corrected the results in supplementary table S9, as well as throughout the text. After correcting the issue with effect and non-effect allele, the sign of the effect (b, and z in table S9a,b), and the sign of the MR Egger intercept (intercept in table S9a,b) of MR results reversed between CP/CPPS - BPH, and BPH- CP/CPPS. The p-values and standard error did not change. A positive direction of effect is more consistent with expectation, given that CP/CPPS and BPH are highly correlated. The effect allele and non-effect allele were coded correctly in CP/CPPS and PCa, so no changes were needed to the MR results for this pair of phenotypes.

We have replaced the line mentioned in the comment with: "Mendelian randomization (MR) was used to evaluate the potential causal interplay between CP/CPPS, BPH, and CaP, since these phenotypes often appear together. "

We hope that by correcting the mistake, and now demonstrating that CP/CPPS and BPH have consistent positive associations clears up the previous version. We have updated the results text accordingly.

We removed the lines "The findings indicate that BPH may be causative of CP/CPPS, suggesting that therapeutic interventions targeting BPH may also potentially alleviate symptoms of CP/CPPS. Alternatively, CP/CPPS may be an extreme phenotype of BPH."

We have corrected the mistake leading to incorrect direction of effect, and now demonstrate that CP/CPPS and BPH have consistent positive associations, which is a more intuitive. We have updated the results text accordingly. We thank the reviewer for identifying this issue before publication of results.

Reviewer 2

- 1) **REVIEWER COMMENT:** All comments have been Addressed

AUTHOR RESPONSE: Thank you.

Reviewer 3

- 1) **REVIEWER COMMENT:** It would be beneficial to include the X chromosome in the analyses.

AUTHOR RESPONSE: Thank you for suggesting that we include the X chromosome in our analyses. MVP continuously grows and updates genetic data, and we incorporate new data as it is available. Although the data are presently accessible within the MVP framework, it was not available when these analyses were performed. We plan to include the X chromosome in future analyses, which we anticipate will yield valuable new insights. Integrating X chromosome data into this study would require a reanalysis of all results, as they all hinge on the GWAS summary statistics that would change with the inclusion of the X chromosome. We appreciate your understanding and believe that the findings presented

here make a valuable contribution to the field, and we look forward to expanding on these results.

- 2) **REVIEWER COMMENT:** To further the authors' point of the shared genetic architecture between CP/CPPS and other diseases, it could be beneficial to conduct a PheWAS of the identified loci (potentially just the clumped and pruned variants utilized in the MR) to investigate the other associated phenotypes.

AUTHOR RESPONSE: As the reviewer suggests, we have now included a PheWAS of the 6 independent significant SNPs from the 3 GWS loci and included any other traits that were significant from the GWAS catalog as a supplemental table (Supplemental Table 6).

REVIEWER RESPONSE: Based on written results and Supplemental Table 6, it would be more apt to remove the term PheWAS and continue with GWAS catalog trait lookup.

AUTHOR RESPONSE: We have revised the wording to remove the term 'PheWAS' in both the main text and the supplemental materials including Supplemental Table 6.

- 3) **REVIEWER COMMENT:** While age was not included in the model, it would be worthwhile to conduct a sensitivity analysis including age as a covariate.

AUTHOR RESPONSE: Conducting the analyses with age at MVP enrollment as a covariate yielded the same three genome-wide significant loci and lead SNPs as our original analysis. The genetic correlation between the EUR CP/CPPS GWAS without age and the EUR CP/CPPS GWAS with age was 0.9922 (SE=0.181). We report these findings in the Results section.

REVIEWER RESPONSE: While I appreciate the sensitivity analysis; I am curious why they used age and MVP enrollment and not age at first CP/CPPS diagnosis.

AUTHOR RESPONSE: Age at MVP enrollment was collected for all participants, ensuring consistent data for analysis. Age at first CP/CPPS diagnosis is unavailable for controls, crucial for GWAS. Thus, we used enrollment age as a covariate.

- 4) **REVIEWER COMMENT:** As the authors use the CC-GWAS method for CaP and BPH, it would be worthwhile to include the MR intercepts as further support of potential pleiotropy between the phenotypes.

AUTHOR RESPONSE: We have added the MR Egger intercepts (all non-significant) to the results section.

REVIEWER RESPONSE: I am satisfied with the inclusion of the MR Egger intercepts as well as the language added in the manuscript.

AUTHOR RESPONSE: Thank you.

- 5) **REVIEWER COMMENT:** While informative, Figure 1B is difficult to read at the presented font size.

AUTHOR RESPONSE: We have increased the font size in Figure 1B to improve readability.

REVIEWER RESPONSE: I am satisfied with the updated title.

AUTHOR RESPONSE: Thank you.

- 6) **REVIEWER COMMENT:** Line 432: Authors state 1,466 traits while the methods state 823 traits.

AUTHOR RESPONSE: We have clarified in the methods that phenotypes were restricted to those with sufficient heritability with the statement: "To ensure robust findings, we restricted our analyses from the 1437 available phenotypes to those displaying a SNP heritability z-score greater than 4, yielding 823 phenotypes for evaluation."

REVIEWER RESPONSE: I am satisfied with the edited wording.

AUTHOR RESPONSE: Thank you.

- 7) **REVIEWER COMMENT:** Line 455 – 458: If the authors could clarify this section. Based on the authors description of the exposure (CP/CPPS) and outcome (BPH) and result, it would seem that CP/CPPS may be causative of BPH, however, it is written as BPH may be causative of CP/CPPS.

AUTHOR RESPONSE: We appreciate the reviewer's request for clarification. We have clarified the following points in our methods, results, and discussion. We explored the causal relationships between CP/CPPS and BPH using bidirectional Mendelian randomization (MR). Our primary analysis modeled CP/CPPS as the exposure and BPH as the outcome, revealing a significant inverse relationship in several MR methods. These findings suggest that CP/CPPS may indeed be causative of BPH.

However, to fully explore the relationship, we also conducted a reverse MR analysis, modeling BPH as the exposure and CP/CPPS as the outcome. This analysis yielded significant results, suggesting that BPH may contribute to the development of CP/CPPS. These bidirectional results highlight a complex interplay between CP/CPPS and BPH and suggest that therapeutic interventions targeting BPH might alleviate symptoms of CP/CPPS, and conversely, managing CP/CPPS might influence the risk of BPH. We acknowledge that this bidirectional relationship might seem confusing and have clarified it in the manuscript to ensure that both directions of causality are communicated. We have revised the relevant sections to emphasize that while CP/CPPS appears to reduce the risk of BPH based on the initial analysis, the reverse relationship also holds true based on our additional findings.

As we describe in our methods and results, we have also included CASUE analyses that employ full GWAS summary statistics to test MR models of causality. Our results were consistent across methods.

REVIEWER RESPONSE: I would recommend adding both the term 'bidirectional' as well as a sentence on using BPH as the exposure and CP/CPPS as the outcome into the mendelian randomization section of the Methods. Otherwise, I am happy with the clarification.

AUTHOR RESPONSE: We rephrased the methods paragraph to clarify our process. The paragraph now reads "Mendelian randomization (MR) was performed using the R package associated with MRbase, TwoSampleMR. We modeled CP/CPPS as the exposure variable

and BPH and CaP as the outcome variables. To evaluate potential bidirectional relationships, we also conducted MR analyses using BPH and CaP as the exposure and CP/CPSP as the outcome."

- 8) REVIEWER COMMENT:** While the title includes prostatitis, suggesting a study of men, it should include the denotation of including only men in the study following the Nature Reporting Summary.

AUTHOR RESPONSE: The title now reads: "The Genetic Basis of Chronic Prostatitis/Chronic Pelvic Pain Syndrome: Insights from a Large-Scale Multi-ancestry Study of Men."

REVIEWER RESPONSE: I am satisfied with the updated title.

AUTHOR RESPONSE: Thank you.

REVIEWER 1

1 REVIEWER COMMENT: Abstract:

We identified the genetic correlations...

Word choice problem: the genetic correlations were estimated. Identified gives the impression they have a fixed value and that the study was able to find those values. That is misleading.

AUTHOR RESPONSE: Thank you, we have adjusted our text. The abstract now reads "We estimated the genetic correlations between CP/CPPS and 12 phenotypes, including prostate cancer (CaP), benign prostatic hyperplasia (BPH), genitourinary disease, abdominal and pelvic pain, back pain, depression, and anxiety."

2 REVIEWER COMMENT: New text on p. 8:

Did all the authors read this? SNPs on high LD ($r^2 \geq 0.001$) with a more significant SNP with a 1Mb window would be pruned. On? Would – under what circumstances?

AUTHOR RESPONSE: The phrase now reads " SNPs in high LD ($r^2 \geq 0.001$) with a more significant SNP within a 1Mb window were pruned. "

3 REVIEWER COMMENT: New text results first line: lifetime phenotypic prevalence
This seems to imply the veterans were all followed until death. Is this true? If not, then the wording used here is quite confusing.

AUTHOR RESPONSE: We have rephrased this sentence to read "Phenotypic prevalence over the available observation period in MVP was 2.23% (22,290/962,151) for CP/CPPS....."

4 REVIEWER COMMENT: Reviewer comment 5:

Results section: "Among Veterans with CP/CPPS, 76.26% had BPH"

Given that over 75% percent of those with CP/CPPS are the same individuals as those with BPH, it seems it is not surprising that they share causal variants (Figure 2)? It is not surprising that genetic correlation is high also. While the results section has been updated to present data, this new data has not been used to provide interpretation in relation to this new finding.

Discussion: The strong genetic correlation between CP/CPPS and BPH, and the GWAS catalog associations, suggest that the genetic causes may be very similar for these conditions.

Isn't high sample overlap a more straightforward explanation for the high genetic correlation?

Also... "suggesting that these traits are deeply linked."

Many who have CP/CPPS also have BPH.

AUTHOR RESPONSE: We thank the reviewer for this comment regarding the interpretation of the relationship between CP/CPPS and BPH. To address this, we have revised the Results section to more explicitly highlight that, although there is substantial comorbidity between CP/CPPS and BPH, the phenotypic overlap is far from complete. The revised Results paragraph now reads:

“Phenotypic prevalence over the available observation period in MVP was 2.23% (22,290/962,151) for CP/CPPS, 3.30% (31,715/962,151) for at CaP, and 31.8% (306,799 / 962,151) for BPH. Among Veterans with CP/CPPS, 76.26% also had BPH, and 7.59% had CaP. Although many individuals with CP/CPPS also had BPH, most individuals with BPH did not have CP/CPPS. Specifically, among individuals with BPH, only 5.5% had CP/CPPS, and 4.2% had CaP. Among CaP cases, 5.3% had CP/CPPS, and 40.6% had BPH. Tetrachoric correlations showed positive associations, strongest between CP/CPPS and BPH ($r_{tet} = 0.47$ [95% CI = 0.47 – 0.47, $p < .001$]), followed by CP/CPPS and CaP ($r_{tet} = 0.18$ [95% CI = 0.18 – 0.18, $p < .001$]), and CaP and BPH ($r_{tet} = 0.11$ [95% CI = 0.10 – 0.11, $p < .001$]).”

This revision quantifies that, while 76% of individuals with CP/CPPS also have BPH, only 5.5% of individuals with BPH have CP/CPPS, and the phenotypic correlation between these conditions is moderate rather than strong ($r = 0.47$). We have also revised the Discussion section to more clearly integrate this point regarding the interplay between phenotypic comorbidity and genetic architecture. The updated Discussion paragraph reads:

“Additionally, MR showed a bidirectional relationship between CP/CPPS and BPH. However, this pattern may reflect shared genetic etiology or pleiotropy rather than strictly causal effects in both directions. Several possible scenarios may underlie these results, including common genetic factors influencing both conditions, pleiotropy where the variants influence both CP/CPPS and BPH through distinct biological pathways, methodological factors like phenotype heterogeneity, true bidirectional causality, or some combination. Consistent with our other results, the finding suggests that factors contributing to the development of one condition might also increase the risk of the other and vice versa, underscoring the need to improve our understanding of the biological mechanisms and pathways shared between these conditions. While clinical comorbidity between CP/CPPS and BPH may contribute to the observed relationship, the moderate phenotypic correlation in our sample, together with genetic findings from two independent approaches (LDSC-derived correlations and twin biometric modeling) support the presence of shared genetic architecture between these two conditions, potentially involving inflammatory processes and endocrine changes. ”

We believe these revisions distinguish the contributions of clinical comorbidity from the evidence supporting shared genetic factors.

- 5 **REVIEWER COMMENT:** New text; p.9: MVP analyses showed lifetime phenotypic prevalence of CP/CPPS at 2.23% (22,290/962,151), CaP at 3.30% (31,715/962,151), and BPH at 2.32% (22,290/962,151). There is an issue since the percent and count of CP/CPPS and BPH are identical. The expectation is that the number of individuals with BPH would be higher than the number with CP/CPPS.

AUTHOR RESPONSE: We thank the reviewer for noting this issue. We have clarified the reported prevalence of BPH in the revised manuscript. The prevalence of BPH is

now correctly stated as 31.8% (306,799/962,151), consistent with expectations based on the study population. The Results section reads "Phenotypic prevalence over the available observation period in MVP was 2.23% (22,290/962,151) for CP/CPPS, 3.30% (31,715/962,151) for at CaP, and 31.8% (306,799 / 962,151) for BPH. "

- 6 **REVIEWER COMMENT:** An error present in a previous version of the manuscript seems like it was not corrected on p.11:
GWAS of Chronic Prostatitis/Chronic Pelvic Pain Syndrome had a regulomeDB score of 3a or higher, indicating possible regulatory relationships. (Table S3).
à lower regulomeDB scores are indicative of regulatory relationships.

AUTHOR RESPONSE: We have corrected this to read "... had a regulomeDB score of 3a or lower, indicating possible regulatory relationships (Table S3)."

- 7 **REVIEWER COMMENT:** Figure 1 legend: I don't see an indication of what S1, S2 and S3 mean. It seems M1-25 are the genes but this is not explicit in the legend.

AUTHOR RESPONSE: We have modified the legend to read "Figure 1: Summary of significant loci (EUR HARE race/ancestry only). A) Manhattan plot showing negative log p-value of the European-only CP/CPPS GWAS. Mapped genes from FUMA are annotated. S1-S3 refer to the genome-wide significant loci ($p < 5E-8$ = red dotted line). M1-M27 refer to the marginal loci ($p < 5E-6$ = yellow dotted line). "

- 8 **REVIEWER COMMENT:** The phrase "collapsing the GWAS signal within the gene set of interest" is not the clearest way to describe what MAGMA does.
Better: to draw from phrases such as: aggregating GWAS signals across all SNPs within each gene; MAGMA incorporates all genes regardless of their statistical significance

AUTHOR RESPONSE: The phrase now reads "TSEA is a method similar to MAGMA's tissue-specific enrichment analysis but is designed for the input of lists of significant genes, whereas MAGMA aggregates GWAS signals across all SNPs within each gene and includes all genes regardless of their statistical significance. "

- 9 **REVIEWER COMMENT:** Under Relationship to BPH and CaP
Better not to use the term: causal interplay. This is vague and suggests some kind of network of causal relationships. Instrumental variables are strictly defined. Each causal arrow is evaluated explicitly. The authors should say what arrows were tested, and then what arrows have associated causal evidence.

AUTHOR RESPONSE: We have replaced "causal interplay" with "causal relationships" (Results, p.15). Specific relationships tested are outlined in the methods (p. Mendelian Randomization) as "We modeled CP/CPPS as the exposure variable and BPH and CaP as the outcome variables. To evaluate potential bidirectional relationships, we also conducted MR analyses using BPH and CaP as the exposure and CP/CPPS as the outcome. " We also report the specific associations in the results and the supplemental table ST9 and ST10

10 **REVIEWER COMMENT:** New text page 16:

“For CP/CPPS against BPH, ... suggesting a causal relationship between CP/CPPS and BPH”. Similarly, For BPH against CP/CPPS, the causal model was a better fit than the null ($p=.01$) and sharing models ($p=.03$), suggesting a causal relationship between CP/CPPS and BPH”.

Here opposite directions of causality are tested (green), and the conclusion by the authors is the same (blue). I am not seeing an interpretation that considers the meaning of the directionality.

AUTHOR RESPONSE: We thank the reviewer for this clarification. While we had acknowledged the bidirectional nature of the MR and CAUSE findings, we have now revised the interpretation to more explicitly consider the implications of directionality. In particular, we clarify that these results may reflect shared etiology or biological feedback rather than independent causal effects in both directions. We have adjusted the text to read:

"CAUSE analyses yielded similar results when comparing causal and sharing models (Table S10). For CP/CPPS against BPH, the causal model was a better fit than both the null ($p=.02$) and sharing models ($p=.01$), suggesting a causal relationship from CP/CPPS to BPH. Similarly, For BPH against CP/CPPS, the causal model was a better fit than the null ($p=.01$) and sharing models ($p=.03$), suggesting a causal relationship in the reverse direction. These bidirectional results likely indicate shared genetic architecture or pleiotropy rather than distinct causal mechanisms in both directions. When CAUSE was applied to CP/CPPS and CaP, neither the sharing nor the causal model was significant, suggesting that horizontal pleiotropy may explain the association between these two phenotypes. Various models of comorbidity can exist, and it is possible that causality operates in both directions, necessitating more research to understand these relationships fully."

REVIEWER 3

REVIEWER COMMENT: Reviewer 3

1. It seems the authors could have performed a GWA scan of the X chromosome among men and provided a Manhattan plot with those findings? The X chromosome test statistics could simply be appended to those for the autosomes. It is understandable that downstream analyses would not be conducted for now on X chromosome data, but the Manhattan plot would at least provide basic findings.

AUTHOR RESPONSE: We have conducted a GWAS of the X chromosome among men and now include these results as Figure 1A, which presents a Manhattan plot displaying association statistics for both the autosomes and the X chromosome. We included genome-wide significant X-linked loci in Table 2. We also revised the Methods and Limitations sections to clarify that post-GWAS analyses were limited to autosomal SNPs, as most tools and external summary statistics currently exclude the X chromosome. Additionally, we updated the Discussion to describe the X-linked findings from the multi-ancestry meta-analysis and highlight the potential relevance of genes such as *GRPR*, *REPS2*, and *RAI2*. These revisions incorporate the X chromosome into

our primary analyses while maintaining transparency about its exclusion from downstream analyses.

REVIEWER 4

- 1 **REVIEWER COMMENT:** I was invited to review this manuscript specifically to comment on the MR analyses. It appears that there was an error in previously submitted analyses, but this has now been resolved. Overall, the MR analyses are performed somewhat algorithmically and uncritically, but also clearly. When I see bidirectional results from an MR study, I think this is likely to represent shared aetiology (i.e. common causes of both traits) rather than literal causal effects in both directions. However, this is a matter of interpretation. Overall, the MR analyses are not a barrier to the publication of this manuscript. As for the rest of the work, these sort of untargeted "throw everything at the wall" analyses aren't my cup of tea if I'm being honest, but I'm sure some people like them. But again, that is a matter of preference.

AUTHOR RESPONSE: We thank the reviewer for their careful review of the MR analyses. We agree that interpreting bidirectional MR findings is complex and that shared etiology is a plausible explanation for the associations observed, in addition to potential causal relationships. In response, we have revised the Discussion to acknowledge this interpretation more explicitly and to present a more cautious interpretation of the bidirectional MR results.

- 2 **REVIEWER COMMENT:** Minor comment: "HARE ensures accurate classification using GIA..." - I doubt any statistical method ensures anything - 'ensure' seems too strong here.

AUTHOR RESPONSE: We have adjusted our text in line with reviewer's suggestion. The phrase now reads "HARE improves the accuracy of classification using GIA to refine and, if necessary, impute SIRE, improving reliability of race/ethnicity assignment in genetic research."

REVIEWER 1

- 1 REVIEWER COMMENT:** The authors have provided adequate responses to the points raised. However, some of the issues, such as the incorrect use of verb tenses or the data reporting error noted under point 5, suggest a need for greater attention to detail. These types of errors may occur elsewhere in the manuscript and ideally the authors should conduct a more comprehensive check to ensure the overall consistency and accuracy of their work.

AUTHOR RESPONSE: We thank the reviewer for their feedback, which has improved the manuscript.